# Source Apportionment of Fine Organic Carbon at an Urban Site of Beijing using a Chemical Mass Balance Model

**Jingsha Xu[1], Di Liu[1,2], Xuefang Wu[1,3], Tuan V. Vu[1,4], Yanli Zhang[5], Pingqing Fu[6], Yele Sun[7], Weiqi Xu[7], Bo Zheng[8,9], Roy M. Harrison[1,*], Zongbo Shi[1,*]**

School of Geography Earth and Environmental Science, University of Birmingham, Birmingham, B15 2TT, UK
Now at: Institute of Atmospheric Physics, Chinese Academy of Sciences, Beijing, 100029, China
School of Geology and Mineral Resources, China University of Geosciences Xueyuan Road 29, Beijing, 100083, China
Now at: Faculty of Life Sciences & Medicine, King's College London, London, WC2R 2LS, UK
Guangzhou Institute of Geochemistry, Chinese Academy of Sciences, Guangzhou, 510640, China
Institute of Surface-Earth System Science, Tianjin University, Tianjin, 300072, China
State Key Laboratory of Atmospheric Boundary Layer Physics and Atmospheric Chemistry, Institute of Atmospheric Physics, Chinese Academy of Sciences, Beijing, 100029, China
State Key Joint Laboratory of Environment Simulation and Pollution Control, School of Environment, Tsinghua University, Beijing, 100084, China
Now at: Laboratoire des Sciences du Climat et de l'Environnement, CEA-CNRS-UVSQ, UMR8212, Gif-sur-Yvette, France

*Correspondence: Zongbo Shi (Z.Shi@bham.ac.uk) and Roy Harrison (r.m.harrison@bham.ac.uk)

## Abstract

Fine particles were sampled from 9[th] November to 11[th] December 2016 and 22[nd] May to 24[th] June 2017 as part of the Atmospheric Pollution and Human Health in a Chinese megacity (APHH-China) field campaigns in urban Beijing, China. Inorganic ions, trace elements, OC, EC, and organic compounds including biomarkers, hopanes, PAHs, n-alkanes and fatty acids, were determined for source apportionment in this study. Carbonaceous components contributed on average 47.2% and 35.2% of total reconstructed $PM_{2.5}$ during the winter and summer campaigns, respectively. Secondary inorganic ions (sulfate, nitrate, ammonium- SNA) accounted for 35.0% and 45.2% of total $PM_{2.5}$ in winter and summer. Other components including inorganic ions ($K^+$, $Na^+$, $Cl^-$), geological minerals, and trace metals only contributed 13.2% and 12.4% of $PM_{2.5}$ during the winter and summer campaigns. Fine OC was explained by seven primary sources (industrial/residential coal burning, biomass burning, gasoline/diesel vehicles, cooking and vegetative detritus) based on a chemical mass balance (CMB) receptor model. It explained an average of 75.7% and 56.1% of fine OC in winter and summer, respectively. Other (unexplained) OC was compared with the secondary OC (SOC) estimated by the EC-tracer method, with correlation coefficients ($R^2$) of 0.58 and 0.73, and slopes of 1.16 and 0.80 in winter and summer, respectively. This suggests that the unexplained OC by CMB was mostly associated with SOC. $PM_{2.5}$ apportioned by CMB showed that the SNA and secondary organic matter were the highest two contributors to $PM_{2.5}$. After these, coal combustion and biomass burning were also significant sources of $PM_{2.5}$ in winter. The CMB results were also compared with results from Positive Matrix Factorization (PMF) analysis of co-located Aerosol Mass Spectrometer (AMS) data. The CMB was found to resolve more primary OA sources than AMS-PMF but the latter could apportion secondary OA sources. The AMS-PMF results for major components, such as coal combustion OC and oxidized OC correlated well with the results from CMB. However, discrepancies and poor agreements were found for other OC sources, such as biomass burning and cooking, some of which were not identified in AMS-PMF factors.

51
52     **Keywords:** $PM_{2.5}$, Beijing, mass closure, CMB, AMS-PMF, source apportionment

## 1 Introduction

Beijing is the capital of China and a hotspot of particulate matter pollution. It has been experiencing severe $PM_{2.5}$ (particulate matter with an aerodynamic diameter of $\leq 2.5\mu m$) pollution in recent decades, as a result of rapid urbanization and industrialization, and increasing energy consumption (Wang et al., 2009). High $PM_{2.5}$ pollution from Beijing could have significant impact on human health (Song et al., 2006a; Li et al., 2013). A case study in Beijing revealed that a 10 $\mu g$ $m^{-3}$ increase of ambient $PM_{2.5}$ concentration will correspondingly increase 0.78%, 0.85% and 0.75% of the daily mortality of the circulatory diseases, cardiovascular diseases and cerebrovascular diseases, respectively (Dong et al., 2013). Furthermore, $PM_{2.5}$ causes visibility deterioration in Beijing. A better understanding of $PM_{2.5}$ sources in Beijing is essential, as it can provide important scientific evidence to develop measures to control $PM_{2.5}$ pollution.

Many studies have identified the possible sources of fine particulate matter in Beijing using various methods (Zheng et al., 2005; Song et al., 2006a; Song et al., 2006b; Li et al., 2015; Zhang et al., 2013; Yu and Wang, 2013). Song et al. (2006a) applied two eigenvector models, principal component analysis/absolute principal component scores (PCA/APCS) and UNMIX to study the sources of $PM_{2.5}$ in Beijing. Some studies used elemental tracers to do source apportionment of $PM_{2.5}$ by applying positive matrix factorization (PMF) (Song et al., 2006b; Li et al., 2015; Zhang et al., 2013; Yu and Wang, 2013). This approach has some underlying challenges. For example, PMF requires a relatively large sample size and a "best" solution of achieved factors requires critical assessment of its mathematical parameters and evaluation of the physical reasonability of the factor profiles (de Miranda et al., 2018; Ikemori et al., 2021; Oduber et al., 2021); secondly, many important $PM_{2.5}$ emission sources do not have a unique elemental composition. Hence, an elemental tracer-based method cannot distinguish sources such as cooking or vehicle exhaust, as they emit mainly carbonaceous compounds (Wang et al., 2009). Generally, organic matter (OM) is composed of primary organic matter (POM) and secondary organic matter (SOM). POM is directly emitted and SOM is formed through chemical oxidation of volatile organic compounds (VOCs) (Yang et al., 2016). OM was the largest contributor to $PM_{2.5}$ mass, which was reported to account for 30%-50% of $PM_{2.5}$ in some Chinese cities such as Beijing, Guangzhou, Xi'an and Shanghai (Song et al., 2007; He et al., 2001; Huang et al., 2014), and can contribute up to 90% of submicron PM mass in Beijing (Zhou et al., 2018). Furthermore, many organic tracers are more specific to particular sources, making them more suitable to identify and quantify different source contributions to carbonaceous aerosols and $PM_{2.5}$.

Chemical Mass balance (CMB) model has been used for source apportionment of PM worldwide, including in the US (Antony Chen et al., 2010), UK (Yin et al., 2015), and China (Chen et al., 2015b). The CMB model assumes that source profiles remain unchanged between the emitter and receptor (Sarnat et al., 2008; Viana et al., 2008). The good performance of CMB and its comparability with other receptor modelling techniques was demonstrated in an intercomparison exercise conducted in Beijing (Xu et al., 2021). A few studies also have applied a CMB model for source apportionment of PM in Beijing (Zheng et al., 2005; Liu et al., 2016; Guo et al., 2013; Wang et al., 2009). For example, Zheng et al. (2005) investigated sources of $PM_{2.5}$ in Beijing, but

the source profiles they used were mainly derived in the United States, which were less
representative of the local sources. Liu et al. (2016) and Guo et al. (2013) apportioned
the sources of $PM_{2.5}$ in a typical haze episode in winter 2013 in Beijing during the
Olympic Games period in summer 2008, respectively. Wang et al. (2009) apportioned
the sources of $PM_{2.5}$ in both winter and summer. A major challenge of the CMB model
is that it cannot quantify the contributions of secondary organic aerosol and unknown
sources, which are often lumped as "unexplained OC".
In this study, $PM_{2.5}$ samples were collected at an urban site of Beijing in winter 2016
and summer 2017. OC, EC, PAHs, alkanes, hopanes, fatty acids and monosaccharide
anhydrides in the $PM_{2.5}$ samples were determined, and applied in the CMB model for
apportioning the organic carbon sources. To ensure that the source profiles used in the
CMB model are representative, we mainly selected data which had been determined in
China. The objectives of this study are: 1) to quantify the contributions of pollution
sources to OC by applying a CMB model and compare them with those at a rural site
of Beijing; 2) to compare the source apportionment results by CMB with those from
Aerosol Mass Spectrometer-PMF analysis (AMS-PMF), to improve our understanding
of different sources of OC.
**2  Methodology**
**2.1 Aerosol sampling**
$PM_{2.5}$ was collected at an urban sampling site (116.39E, 39.98N) - the Institute of
Atmospheric Physics (IAP) of the Chinese Academy of Sciences in Beijing, China from
9[th] November to 11[th] December 2016 and 22[nd] May to 24[th] June 2017, as part of the
Atmospheric Pollution and Human Health in a Chinese megacity (APHH-China) field
campaigns (Shi et al., 2019). The sampling site (Fig. 1) is located in the middle between
the North 3[rd] Ring Road and North 4[th] Ring Road and approximately 200 m from a
major highway. Hence, it is subject to many local sources, such as traffic, cooking, etc.
The location of a rural site in Beijing - Pinggu during the APHH-China campaigns is
also shown in Fig. 1. The rural site in Xibaidian village in Pinggu is about 60 km away
from IAP and 4 km north-west of the Pinggu town centre. It is surrounded by trees and
farmland with several similar small villages nearby. A provincial highway is
approximately 500 m away on its eastside running north-south. This site is far from
industrial sources and located in a residential area. Other information regarding the
sampling site is described elsewhere (Shi et al., 2019).

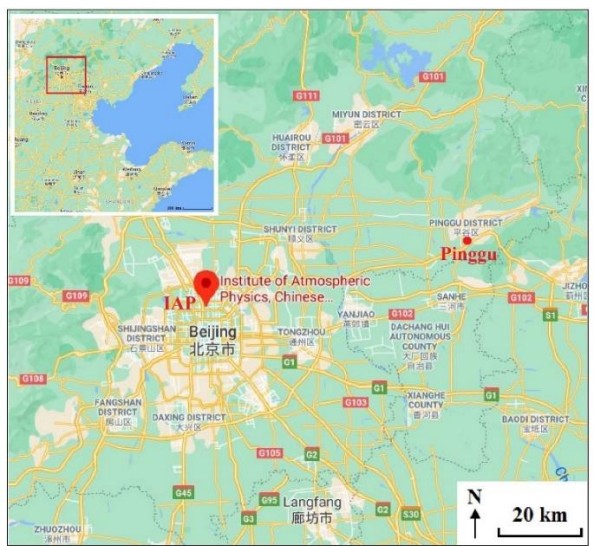

**Figure 1.** Locations of the sampling sites in Beijing (IAP - urban site: Institute of Atmospheric Physics of the Chinese Academy of Sciences; Pinggu - rural site) (source: © Google Maps).

PM$_{2.5}$ samples were collected on pre-baked (450°C for 6h) large quartz filters (Pallflex, 8×10 inch) by Hi-Vol air sampler (Tisch, USA) at a flow rate of 1.1 m$^3$ min$^{-1}$. A Medium-Vol air sampler (Thermo Scientific Partisol 2025i) was also deployed at the same location to collect PM$_{2.5}$ samples simultaneously on 47 mm PTFE filters at a flow rate of 15.0 L min$^{-1}$. Field blanks were also collected with the pump turned off during the sampling campaign. Before and after sampling, all filters were put in a balance room and equilibrated at a constant temperature and relative humidity (RH) for 24h prior to any gravimetric measurements, which were 22°C and 30% RH for summer samples, 21°C and 33% RH for winter samples. PM$_{2.5}$ mass was determined through the weighing of PTFE filters using a microbalance (Sartorius model MC5, precision: 1 µg). After that, filters were wrapped separately with aluminum foil and stored at under −20°C in darkness until analysis. The large quartz filters were analyzed for OC, EC, organic compounds and ion species, while small PTFE filters were used for the determination of PM$_{2.5}$ mass and metals. Online PM$_{2.5}$ were determined by the TEOM FDMS 1405-DF instrument at IAP with filter equilibrating and weighing conditions comparable with the United States Federal Reference Method (RH: 30-40%; temperature; 20-23°C) (Le et al., 2020; U.S.EPA, 2016).

**2.2 Chemical Analysis**

**2.2.1 OC and EC**

A 1.5 cm$^2$ punch from each large quartz filter sample was taken for organic carbon (OC) and elemental carbon (EC) measurements by a thermal/optical carbon analyzer (model RT-4, Sunset Laboratory Inc., USA) based on the EUSAAR2 (European Supersites for Atmospheric Aerosol Research) transmittance protocol (Cavalli et al., 2010; Chen et al., 2015a). Replicate analyses of OC and EC were conducted once every ten samples. The uncertainties from duplicate analyses of filters were <10%. All sample results were corrected by the values obtained from field blanks, which were 0.40 and 0.01 µg m$^{-3}$ for OC and EC, respectively. Details of the OC/EC measurement method can be found elsewhere (Paraskevopoulou et al., 2014). The instrumental limits of detection of OC

and EC in this study were estimated to be 0.03 and 0.05 µg m$^{-3}$, respectively.

### 2.2.2 Organic compounds

Organic tracers, including 11 n-alkanes (C$_{24}$-C$_{34}$), 2 hopanes (17a (H) -22, 29, 30-
Trisnorhopane, 17b (H), 21a (H) -Norhopane), 17 PAHs (retene, phenanthrene,
anthracene, fluoranthene, pyrene, benz(a)anthracene, chrysene, benzo(b)fluoranthene,
benzo(k)fluoranthene, benzo(e)pyrene, benzo(a)pyrene, perylene, Indeno(1,2,3-
cd)pyrene, dibenz(a,h)anthracene, benzo(ghi)perylene, coronene, picene), 3
anhydrosugars (levoglucosan, mannosan, galactosan), 2 fatty acids (palmitic acid,
stearic acid) and cholesterol in the PM$_{2.5}$ samples were determined in this study. 9 cm$^2$
of the large quartz filters were extracted 3 times with dichloromethane/methanol (HPLC
grade, v/v: 2:1) under ultrasonication for 10 minutes. The extracts were then filtered
and concentrated using a rotary evaporator under vacuum, and blown down to dryness
with pure nitrogen gas. 50 µL of N,O-bis-(trimethylsilyl)trifluoroacetamide (BSTFA)
with 1% trimethylsilyl chloride and 10 µL of pyridine were then added to the extracts,
which were left reacting at 70 °C for 3 h to derivatize -COOH to TMS esters and -OH
to TMS ethers. After cooling to room temperature, the derivatives were diluted with
140 µL of internal standards (C13 n-alkane, 1.43 ng µL$^{-1}$) in n-hexane prior to GC-MS
analysis. The final solutions were analyzed by a gas chromatography mass spectrometry
system (GC/MS, Agilent 7890A GC plus 5975C mass-selective detector) fitted with a
DB-5MS column (30 m × 0.25 mm × 0.25 µm). The GC temperature program and MS
detection details were reported in Li et al. (2018). Individual compounds were identified
through the comparison of mass spectra with those of authentic standards or literature
data (Fu et al., 2016). Recoveries for these compounds were in a range of 70-100%,
which were obtained by spiking standards to pre-baked blank quartz filters followed by
the same extraction and derivatization procedures. Field blank filters were analyzed the
same way as samples for quality assurance, but no target compounds were detected.

### 2.2.3 Inorganic components

Half of the PTFE filter was extracted with 10 mL ultrapure water for the analysis of
inorganic ions. Major inorganic ions including Na$^+$, K$^+$, NH$_4^+$, Cl$^-$, NO$_3^-$ and SO$_4^{2-}$ were
determined by using an ion chromatograph (IC, Dionex, Sunnyvale, CA, USA), the
detection limits (DLs) of them were 0.032, 0.010, 0.011, 0.076, 0.138, 0.240 and 0.142
µg m$^{-3}$ respectively. The analytical uncertainty was less than 5% for all inorganic ions.
An intercomparison study showed that our IC analysis of the above-mentioned ions
agreed well with those of the other laboratories (Xu et al., 2020). Trace metal including
Al (DLs in µg m$^{-3}$, 0.221), Si (0.040), Ca (0.034), Ti (0.003) and Fe (0.044) were
determined by X-ray fluorescence spectrometer (XRF). Other elements including V, Cr,
Co, Mn, Ni, Cu, Zn, As, Sr, Cd, Sb, Ba and Pb were analyzed by Inductively-coupled
plasma-mass spectrometer (ICP-MS) after extraction of 1/2 PTFE filter by diluted acid
mixture (HNO$_3$/HCl), and the detection limits of them were 1.32, 0.25, 0.04, 0.06, 2.05,
1.25, 1.22, 1.74, 0.02, 0.03, 0.11, 0.06 and 0.04 ng m$^{-3}$, respectively. Mass
concentrations of all inorganic ions and elements in this study were corrected for the
field blank values, and the methods were quality assured with standard reference
materials.

## 2.3 Chemical Mass Closure (CMC) Method

A Chemical Mass Closure analysis was carried out, which includes secondary inorganic ions (sulfate, nitrate, ammonium; SNA), sodium, potassium and chloride salts, geological minerals, trace elements, organic matter (OM), EC and bound water in reconstructed $PM_{2.5}$. Geological minerals were calculated applying the equation (Eq. 1) (Chow et al., 2015):

$$Geological\ minerals = 2.2Al + 2.49Si + 1.63Ca + 1.94Ti + 2.42Fe \tag{1}$$

Trace elements were the sum of all analysed elements excluding Al, Si, Ca, Ti and Fe. The average OM/OC ratios of organic aerosols (OA) from AMS elemental analysis were applied to calculate OM, which were 1.75±0.16 and 2.00±0.19 in winter and summer, respectively. Based on the concentrations of inorganic ions and gas-phase $NH_3$, particle bound water was calculated by ISORROPIA II model (available at http://isorropia.eas.gatech.edu) in forward mode and thermodynamically metastable phase state (Fountoukis and Nenes, 2007). Two sets of calculations were done for online and offline data, differing at the temperature and relative humidity as specified above.

## 2.4 Chemical Mass Balance (CMB) model

The chemical mass balance model (US EPA CMB8.2) was applied in this study to apportion the sources of OC by utilizing a linear least squares solution. Both uncertainties in source profiles and ambient measurements were taken into consideration in this model. The source profiles applied here were from local studies in China to better represent the source characteristics, including straw burning (wheat, corn, rice straw burning) (Zhang et al., 2007b), wood burning (Wang et al., 2009), gasoline and diesel vehicles (including motorcycles, light- and heavy-duty gasoline and diesel vehicles) (Cai et al., 2017), industrial and residential coal combustion (including anthracite, sub-bituminite, bituminite, and brown coal) (Zhang et al., 2008), and cooking (Zhao et al., 2015), except vegetative detritus (Rogge et al., 1993; Wang et al., 2009). The source profiles with EC and organic tracers used in the CMB model were provided in Table S1 of Wu et al. (2020). The selected fitting species were EC, levoglucosan, palmitic acid, stearic acid, fluoranthene, phenanthrene, retene, benz(a)anthracene, chrysene, benzo(b)fluoranthene, benzo(k)fluoranthene, benzo[ghi]perylene, picene, 17a (H) -22, 29, 30-trisnorhopane, 17b (H), 21a (H) - norhopane and n-alkanes (C24-C33), the concentrations of which are provided in Table 1. The essential criteria in this model were met to ensure reliable fitting results. For instance, in all samples, $R^2$ were >0.80 (mostly >0.9), $Chi^2$ were <2, $T_{stat}$ values were mostly greater than 2 except the source of vegetative detritus, and C/M ratios (ratio of calculated to measured concentration) for all fitting species were in range of 0.8-1.2 in this study.

## 2.5 Positive Matrix Factorization analysis of data obtained from Aerosol Mass Spectrometer (AMS-PMF)

An Aerodyne AMS with a $PM_1$ aerodynamic lens was deployed on the roof of the neighboring building- the Tower branch of IAP for real-time measurements of non-

refractory (NR) chemical species from $16^{th}$ November to $11^{th}$ December 2016 and $22^{nd}$ May to $24^{th}$ June 2017. The detailed information of the sampling sites is given elsewhere (Xu et al., 2019b). The submicron particles were dried and sampled into the AMS at a flow of ~0.1 L min$^{-1}$. NR-PM$_1$ can be quickly vaporized by the 600 $\circ$C tungsten vaporizer and then the NR-PM$_1$ species including organics, Cl$^-$, NO$_3^-$, SO$_4^{2-}$ and NH$_4^+$ were measured by AMS in mass sensitive V mode (Sun et al., 2020). Details of AMS data analysis, including the analysis of organic aerosol (OA) mass spectra can be found elsewhere (Xu et al., 2019b). The source apportionment of organics in NR-PM$_1$ was carried out by applying PMF to the high-resolution mass spectra of OA, while that of fine OC in this study was conducted by applying source profiles along with an offline chemical speciation dataset. The procedures of the pretreatment of spectral data and error matrices can be found elsewhere (Ulbrich et al., 2009). It is noted that the data were missing during the period $09^{th}$ - $15^{th}$ November 2016 due to the malfunction of the AMS.

## 3 Results and discussion

### 3.1 Characteristics of PM$_{2.5}$ and Carbonaceous Compounds

Mean concentrations of PM$_{2.5}$, OC, EC and organic tracers during wintertime ($9^{th}$ November to $11^{th}$ December 2016) and summertime ($22^{nd}$ May to $24^{th}$ June 2017) at the IAP site are summarized in Table 1 and Fig. S1. The average PM$_{2.5}$ concentration was 94.8±64.4 µg m$^{-3}$ during the whole winter sampling campaign. The winter sampling period was divided into haze (daily PM$_{2.5}$ > 75 µg m$^{-3}$) and non-haze days (<75 µg m$^{-3}$), based on the National Ambient Air Quality Standard Grade II of the limit for 24-hour average PM$_{2.5}$ concentration. The differentiation between haze and non-haze days enabled us to study the major sources contributing to the haze formation. The average daily PM$_{2.5}$ was 136.7±49.8 and 36.7±23.5 µg m$^{-3}$ on haze and non-haze days, respectively. Daily PM$_{2.5}$ in the summer sampling period was 30.2±14.8 µg m$^{-3}$, comparable with that on winter non-haze days.

OC concentrations ranged between 3.9-48.8 µg m$^{-3}$ (mean: 21.5 µg m$^{-3}$) and 1.8-12.7 µg m$^{-3}$ (mean: 6.4 µg m$^{-3}$) during winter and summer, respectively. They are comparable with the OC concentrations in winter (23.7 µg m$^{-3}$) and summer (3.78 µg m$^{-3}$) in Tianjin, China during an almost simultaneous sampling period (Fan et al., 2020), but much lower than the OC concentration (17.1 µg m$^{-3}$) in summer 2007 in Beijing (Yang et al., 2016). The average OC concentration during haze days (29.4±9.2 µg m$^{-3}$) was approximately three times that of non-haze days (10.7±6.2 µg m$^{-3}$) during winter. The average EC concentration during winter was 3.5±2.0 µg m$^{-3}$; its concentration was 4.6±1.3 µg m$^{-3}$ on haze days, approximately 2.4 times that on winter non-haze days (1.9±1.6 µg m$^{-3}$) and 5 times that (0.9±0.4 µg m$^{-3}$) during the summer sampling period. The OC and EC concentrations in this study were comparable with the OC (27.9 ± 23.4 µg m$^{-3}$) and EC (6.6 ± 5.1 µg m$^{-3}$) concentrations in winter Beijing in 2016 (Qi et al., 2018), but much lower than those in an urban area of Beijing during winter (OC and EC: 36.7±19.4 and 15.2±11.1 µg m$^{-3}$) and summer (10.7±3.6 and 5.7±2.9 µg m$^{-3}$) in 2002 (Dan et al., 2004).

On average, OC and EC concentrations in winter were 3.3 and 3.9 times those in summer. Additionally, OC and EC were well-correlated in this study, with $R^2$ values of 0.85 and 0.63 during winter and summer, respectively, suggesting similar paths of OC and EC dispersion and dilution, and/or similar sources of carbonaceous aerosols, especially in winter. Less correlated OC and EC in summer could be a result of SOC formation. SOC in this study was estimated and is discussed in section 3.3.7.

**Table 1.** Summary of measured concentrations at IAP site in winter and summer.

| Compounds[a]/ ng m$^{-3}$ | Winter | | Winter (n=31) | Summer (n=34) |
|---|---|---|---|---|
| | Haze[d] (n=18) | Non-haze[e] (n=13) | | |
| PM$_{2.5}$ (µg m$^{-3}$) | 136.7±49.8 (80.5-239.9)[b] | 36.7±23.5 (10.3-72) | 94.8±64.4 (10.3-239.9) | 30.2±14.8 (12.2-78.8) |
| OC (µg m$^{-3}$) | 29.4±9.2 (13.7-48.8) | 10.7±6.2 (3.9-21.5) | 21.5±12.3 (3.9-48.8) | 6.4±2.3 (1.8-12.7) |
| EC (µg m$^{-3}$) | 4.6±1.3 (1.6-6.6) | 1.9±1.6 (0.3-5.2) | 3.5±2.0 (0.3-6.6) | 0.9±0.4 (0.2-1.7) |
| SOC[c] (µg m$^{-3}$) | 10.3±5.7 (2.9-24.6) | 2.9±1.4 (0.0-5.5) | 7.2±5.7 (0.0-24.6) | 2.3±1.4 (0.0-6.0) |
| Levoglucosan | 348.2±148.0 (83.1-512.5) | 195.0±163.7 (19.1-539.5) | 278.5±171.4 (19.1-539.5) | 26.1±28.3 (2.9-172.2) |
| Palmitic acid | 376.2±234.9 (44.5-1089.6) | 278±280.6 (33.8-1137.2) | 335±255.3 (33.8-1137.2) | 25.2±11.9 (9.4-68) |
| Stearic acid | 207.1±181.4 (23-846.7) | 163.6±228.1 (17.3-903.2) | 188.8±199.8 (17.3-903.2) | 16.0±7.2 (5.6-36.4) |
| Phenanthrene | 8.6±6.1 (1.8-19) | 5.6±6.1 (1-24.8) | 7.3±6.2 (1-24.8) | 0.7±0.7 (0-3.8) |
| Fluoranthene | 25.1±19.6 (4.2-76.2) | 16.1±21.3 (4.2-84.3) | 21.3±20.5 (4.2-84.3) | 0.4±0.2 (0-0.9) |
| Retene | 16±14.9 (2-52.2) | 11.1±12.1 (0.5-45.5) | 13.9±13.8 (0.5-52.2) | 0±0 (0-0.1) |
| Benz(a)anthracene | 21.5±16.5 (0.3-62.7) | 10.8±9.3 (1.4-30.5) | 17±14.8 (0.3-62.7) | 0.2±0.1 (0-0.5) |
| Chrysene | 22.6±14.1 (3.7-47.3) | 13.6±15.6 (0.1-59.5) | 18.8±15.2 (0.1-59.5) | 0.2±0.1 (0-0.3) |
| Benzo(b)fluoranthene | 52.6±29 (10.7-98) | 28.1±31 (2.4-113.6) | 42.3±31.8 (2.4-113.6) | 0.7±0.5 (0-2) |
| Benzo(k)fluoranthene | 12.2±8 (0-25.3) | 6.7±6.8 (0-23.7) | 9.9±7.9 (0-25.3) | 0.2±0.1 (0-0.4) |
| Picene | 0.8±0.8 (0-2.6) | 0.3±0.5 (0-1.3) | 0.6±0.7 (0-2.6) | 0±0 (0-0) |
| Benzo(ghi)perylene | 7.0±4.7 (0-13.6) | 4.0±4.1 (0-14.0) | 5.6±4.6 (0-14.0) | 0±0.1 (0-0.3) |
| 17a (H) -22, 29, 30-Trisnorhopane | 2.7±1.6 (0.6-6.7) | 1.6±1.5 (0.3-6) | 2.2±1.6 (0.3-6.7) | 0±0.1 (0-0.4) |
| 17b (H), 21a (H) - Norhopane | 3.1±1.6 (0.9-6.6) | 1.8±1.8 (0.3-7.3) | 2.6±1.8 (0.3-7.3) | 0±0 (0-0.2) |
| C24 | 26.3±15.3 (7.8-55.5) | 18±19.2 (2.1-71.2) | 22.5±17.4 (2.1-71.2) | 1.4±0.6 (0.5-3.3) |
| C25 | 28.2±15.6 (8.5-59) | 19.5±20.5 (2.3-76.2) | 24.2±18.3 (2.3-76.2) | 2.9±1.5 (0.5-6.5) |
| C26 | 18.9±10.2 (5.8-40.2) | 13±13.1 (1.8-48.2) | 16.2±11.8 (1.8-48.2) | 1.6±0.7 (0.3-4.3) |
| C27 | 20.4±9.2 (6.1-37.1) | 13.8±12.5 (2.2-43.5) | 17.4±11.2 (2.2-43.5) | 4.4±2 (0.6-11.7) |
| C28 | 10.6±4.8 (3.2-19.2) | 6.9±5.7 (1.5-19.3) | 8.9±5.5 (1.5-19.3) | 1.4±0.6 (0.3-2.9) |
| C29 | 22.3±10.1 (5.9-39.7) | 14.3±12.6 (3-39) | 18.7±11.9 (3-39.7) | 5.2±3.3 (0.4-20.7) |
| C30 | 6.8±2.9 (2.2-11.4) | 4.5±3.1 (1-9.7) | 5.7±3.2 (1-11.4) | 1±0.4 (0.2-2) |
| C31 | 11.6±4.2 (3.5-17.7) | 7.7±5.8 (1.2-18.7) | 9.8±5.3 (1.2-18.7) | 4.3±3.2 (0.4-20) |
| C32 | 6.1±2.6 (1.7-9.3) | 3.9±2.6 (0.7-8.2) | 5.1±2.8 (0.7-9.3) | 0.9±0.4 (0.2-1.7) |
| C33 | 5.8±2.7 (1.7-11.5) | 3.9±3.1 (0.9-9.6) | 4.9±3 (0.9-11.5) | 1.8±1.1 (0.1-6.3) |
| C34 | 2.1±2.1 (0-5.5) | 1.2±1.4 (0-4) | 1.7±1.8 (0-5.5) | 0.3±0.3 (0-0.9) |

[a] The unit is ng m$^{-3}$ for all organic compounds and µg m$^{-3}$ for PM$_{2.5}$, OC, EC and SOC; [b] mean±SD (min-max); [c] SOC concentration was calculated by EC-tracer method; [d] Haze days: PM$_{2.5}$≥75 µg m$^{-3}$; [e] Non-haze days: PM$_{2.5}$<75 µg m$^{-3}$;

## 3.2 Chemical Mass Closure (CMC)

The composition of PM$_{2.5}$ applying the chemical mass closure method is plotted in Fig.2 and summarized in Table S1. Because the gravimetrically measured mass (offline PM$_{2.5}$) differs slightly from online PM$_{2.5}$ (Fig. S2), the regression analysis results between mass reconstructed using mass closure (reconstructed PM$_{2.5}$) and both measured PM$_{2.5}$ (offline PM$_{2.5}$/ online PM$_{2.5}$) were investigated and plotted in Fig. 3.

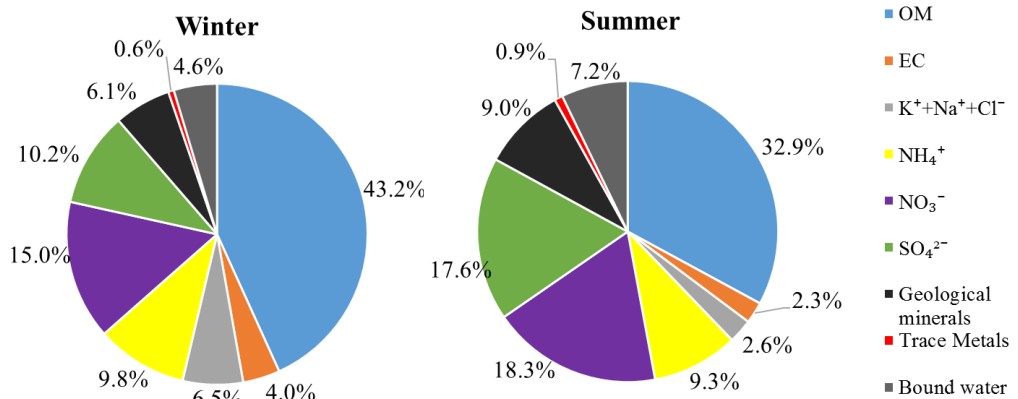

**Figure 2.** Chemical components of reconstructed PM$_{2.5}$ (offline) applying mass closure method.

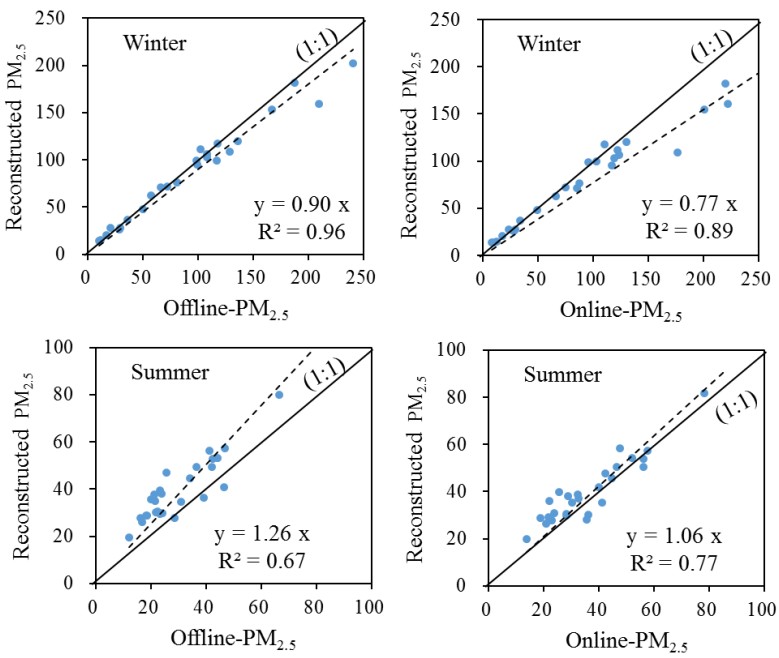

**Figure 3.** Regression results between reconstructed PM$_{2.5}$ and offline/online PM$_{2.5}$ by chemical mass closure method.

As shown in Fig. 3, measured offline/online PM$_{2.5}$ were moderately well correlated with the reconstructed PM$_{2.5}$ with slopes of 0.77~1.26 and R$^2$ of 0.67~0.96. In winter, the regression results were good between reconstructed PM$_{2.5}$ and offline-PM$_{2.5}$. For online-PM$_{2.5}$, it was much higher than the reconstructed PM$_{2.5}$ when the mass was over 170 µg m$^{-3}$. After excluding the outliers (2 outliers of offline-PM$_{2.5}$ > 200 µg m$^{-3}$ and 4 outliers of online-PM$_{2.5}$ > 170 µg m$^{-3}$), the regression results improved with both slopes and R$^2$ approaching unity (Fig. S3). This could indicate some uncertainties in offline and/or online PM$_{2.5}$ measurement for heavily polluted samples, or the applied OM/OC ratio in winter was not suitable for converting OC to OM in heavily polluted samples. During the summer campaign, the slope of the reconstructed PM$_{2.5}$ and online-PM$_{2.5}$ was close to 1, but that of reconstructed PM$_{2.5}$ and offline-PM$_{2.5}$ was 1.26. This could be due to the loss of semi-volatile compounds from PTFE filters or the positive artefacts

of quartz filters for chemical analyses, which can absorb more organics than PTFE filters that are used for PM weighing. To avoid loss of semi-volatiles, all collected samples were stored in cold conditions, including during shipment. The datapoints were more scattered in summer, which could result from the large difference in OM-OC relationships from day to day. The reconstructed inorganics (reconstructed $PM_{2.5}$ excluding OM) correlated well with offline-$PM_{2.5}$, but OM did not (Fig. S4). Hence, the discrepancies of between reconstructed $PM_{2.5}$ and offline/online $PM_{2.5}$ in summer may be mainly attributable to variable OM/OC ratios.

During the winter campaign, the carbonaceous components (OM & EC) accounted for 47.2% of total reconstructed $PM_{2.5}$, followed by the secondary inorganic ions ($NH_4^+$, $SO_4^{2-}$, $NO_3^-$) (35.0%). In summer, on the contrary, secondary inorganic salts represented 45.2% of $PM_{2.5}$ mass, followed by carbonaceous components (35.2%). Bound water contributed 4.6% and 7.2% of $PM_{2.5}$ during the winter and summer, respectively. All other components combined accounted for 13.2% and 12.4% of $PM_{2.5}$ during the winter and summer campaigns, respectively.

## 3.3 Source apportionment of fine OC in urban Beijing applying a CMB model

The CMB model resolved seven primary sources of OC in winter and summer, including vegetative detritus, straw and wood burning (biomass burning, BB), gasoline vehicles, diesel vehicles, industrial coal combustion (Industrial CC), residential coal combustion (Residential CC) and cooking. It explained an average of 75.7% (45.3-91.3%) and 56.1% (34.3-76.3%) of fine OC in winter and summer, respectively. The averaged CMB source apportionment results in winter and summer are presented in Table 2. Daily source contribution estimates to fine OC and the relative abundance of different sources contributions to OC in winter and summer are shown in Fig. 4.

During the winter campaign, coal combustion (industrial and residential CC, 7.5 µg $m^{-3}$, 35.0% of OC) was the most significant contributor to OC, followed by Other OC (5.3 µg $m^{-3}$, 24.8%), biomass (3.8 µg $m^{-3}$, 17.6%), traffic (gasoline and diesel vehicles, 2.6 µg $m^{-3}$, 11.9%), cooking (2.2 µg $m^{-3}$, 10.3%), vegetative detritus (0.09 µg $m^{-3}$, 0.4%). On winter haze days, industrial coal combustion, cooking and Other OC were significantly higher (nearly tripled) compared to non-haze days. During the summer campaign, Other OC (2.9 µg $m^{-3}$, 45.6%) was the most significant contributor to OC, followed by coal combustion (2.0 µg $m^{-3}$, 31.1%), cooking (0.7 µg $m^{-3}$, 10.3%), traffic (0.4 µg $m^{-3}$, 6.1%), biomass burning (0.3 µg $m^{-3}$, 5.3%), and vegetative detritus (0.1 µg $m^{-3}$, 1.7%).

**Table 2.** Source contribution estimates (SCE, µg $m^{-3}$) for fine OC in urban Beijing during winter and summer from the CMB model

| Sources | Winter | | Winter (n=31) | Summer (n=34) |
| --- | --- | --- | --- | --- |
| | Haze (n=18) | Non-haze (n=13) | | |
| Vegetative detritus | 0.11±0.08 | 0.07±0.08 | 0.09±0.08 | 0.11±0.08 |
| Biomass burning | 4.80±2.23 | 2.38±2.57 | 3.78±2.64 | 0.34±0.39 |
| Gasoline vehicles | 2.35±1.27 | 1.59±1.85 | 2.03±1.56 | 0.31±0.16 |
| Diesel vehicles | 0.83±1.43 | 0.14±0.33 | 0.54±1.15 | 0.08±0.16 |
| Industrial coal combustion | 7.09±4.17 | 1.95±1.36 | 4.94±4.15 | 1.82±0.72 |
| Residential coal combustion | 3.64±3.72 | 1.16±0.96 | 2.60±3.12 | 0.18±0.11 |

| | | | | |
|---|---|---|---|---|
| Cooking | 3.23±2.30 | 0.85±0.52 | 2.23±2.13 | 0.66±0.43 |
| Other OC[a] | 7.4±5.6 | 2.5±1.4 | 5.3±4.9 | 2.9±1.5 |
| Calculated OC[b] | 22.0±6.5 | 8.2±5.3 | 16.2±9.1 | 3.5±1.2 |
| Measured OC | 29.4±9.2 | 10.7±6.2 | 21.5±12.3 | 6.4±2.3 |

366    [a] *Other OC is calculated by subtracting calculated OC from measured OC;.*

367    [b] *Calculated OC is the sum of OC from all seven primary sources: vegetative detritus, biomass burning,*

368    *gasoline vehicles, diesel vehicles, industrial coal combustion, residential coal combustion and cooking.*

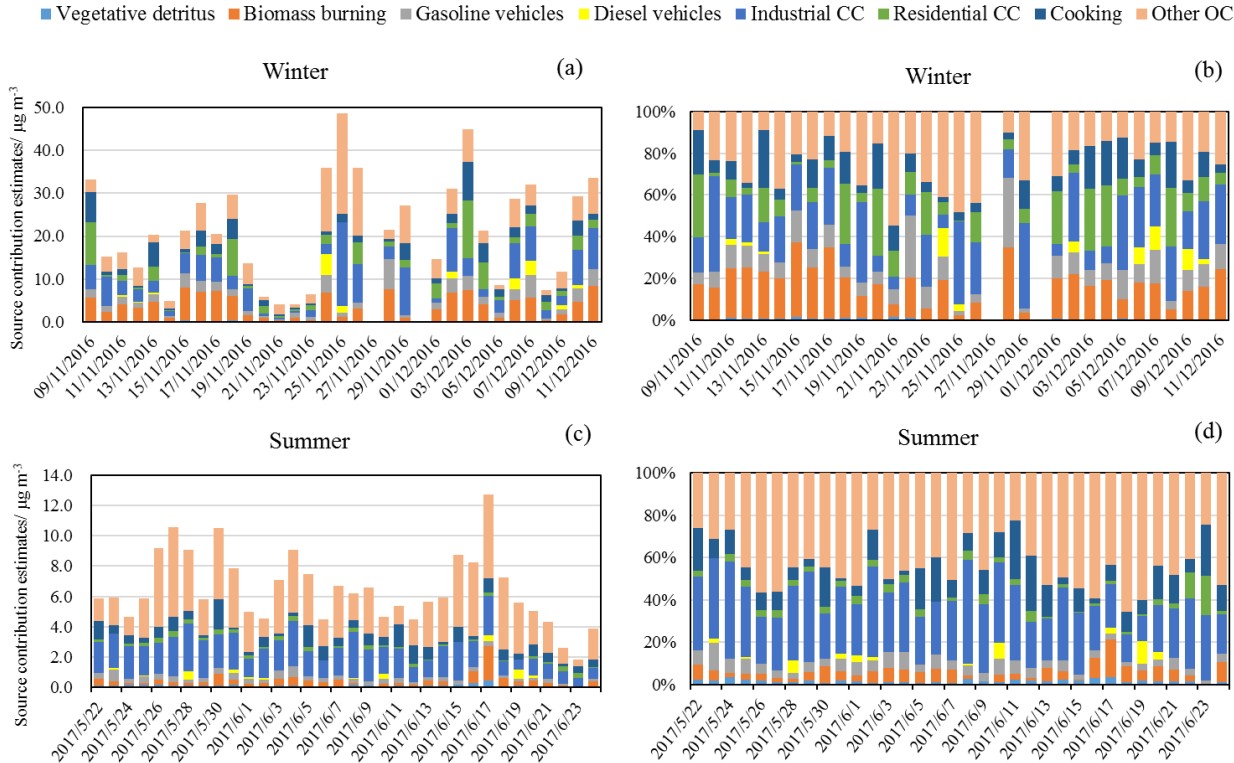

369

370 **Figure 4.** Daily source contribution estimates to fine OC in (a) winter and (c) summer
371 and their relative abundance in winter (b) and summer (d)

### 3.3.1 Industrial and residential coal combustion

In China, a large amount of coal is used in thermal power plant, industries, urban and rural houses in northern China, especially during the heating period (mid-November to mid-March) (Huang et al., 2017; Yu et al., 2019). But urban household coal use experienced a remarkable drop of 58% during 2005-2015, which is much higher than that of rural household coal use (5% of decrease) (Zhao et al., 2018). In this study, coal combustion is the single largest source that contributed to primary OC in both winter and summer. In addition, industrial CC was a more significant source of OC than residential CC in urban Beijing. On average, coal combustion related OC was 7.5±5.0 µg m$^{-3}$ (34.5±9.8% of OC) in winter, which was more than 3 times of that in summer - 2.0±0.8 µg m$^{-3}$ (32.3±10.2% of OC), but the percentage contribution is similar. A similar seasonal trend was also found in other studies in Beijing (Zheng et al., 2005; Wang et al., 2009), but the relative contribution of coal combustion was much lower than in this study. Industrial CC derived OC was 4.94±4.15 and 1.82±0.72 µg m$^{-3}$ in winter and summer, respectively. Residential CC derived OC was 2.60±3.12 and 0.18±0.11 µg m$^{-3}$ in winter and summer, respectively. Residential CC was much higher in winter compared to that in summer. On haze days, industrial CC and residential CC derived OC were 3.6 and 3.1 times that on non-haze days, respectively, indicating an important contribution to haze formation from industrial CC.

Coal combustion is also a major source for particulate chloride (Chen et al., 2014). Because Beijing is an inland city, the contribution of marine aerosols to particulate Cl$^-$ is considered minor, which is also supported by the higher Cl$^-$/Na$^+$ mass ratios in winter (10.1±4.8) and summer (2.7±1.8) than sea water (1.81), indicative of significant contributions from anthropogenic sources (Bondy et al., 2017). Yang et al. (2018) also reported that the contribution of sea-salt aerosol to fine particulate chloride was negligible in China inland areas even during summer. Hence, Cl$^-$ in this study was mainly from anthropogenic sources. The time series of OC from coal combustion (OC-CC) and Cl$^-$ during winter and summer of Beijing are shown in Fig. 5. OC-CC and Cl$^-$ exhibited similar trends in both seasons. The correlation coefficient ($R^2$) between OC-CC and Cl$^-$ during winter was 0.62, which could be attributed to enhanced coal combustion activities in this season. No significant correlation between the two was found during the summer campaign, indicating the abundance of Cl$^-$ in summer was more influenced by other sources, probably including biomass burning. In addition, due to the semi-volatility of ammonium chloride, it is liable to evaporate in summer (Pio and Harrison, 1987). A similar phenomenon has been observed in Delhi (Pant et al., 2015).

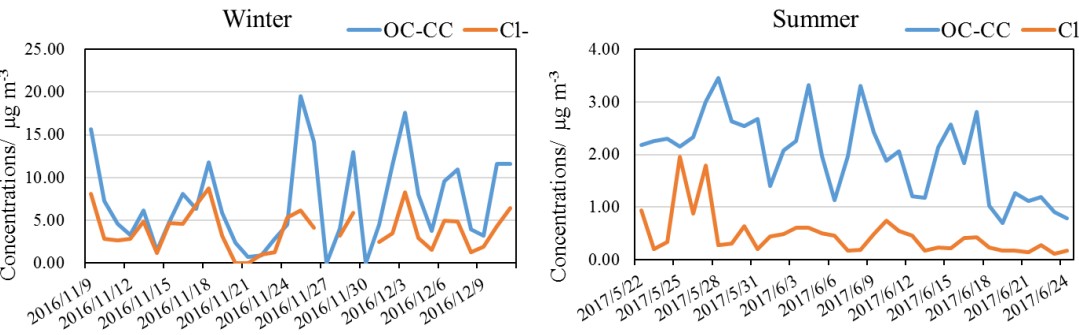

 **Figure 5.** Time series of OC from coal combustion (OC-CC) and Cl⁻ in winter and
summer in Beijing

### 3.3.2 Biomass burning

Biomass burning (BB), including straw and wood burning, is an important source of
atmospheric fine OC, which ranked as the second highest primary source of OC, after
industrial coal combustion during the winter campaign, and third highest during the
summer campaign after industrial CC and cooking. As shown in Fig. 4, the relative
abundance of BB derived-OC during the winter campaign is much higher than the
summer campaign. BB-derived OC from the CMB results was $3.78\pm2.64$ µg m⁻³ and
$0.34\pm0.39$ µg m⁻³ in winter and summer, contributing 17.6% and 5.3% of OC in these
two seasons, respectively. These results are lower than those in 2005-2007 Beijing
when BB accounted for 26% and 11% of OC in winter and summer, respectively (Wang
et al., 2009). The BB-derived OC on winter haze days ($4.80\pm2.23$ µg m⁻³) was
approximately double that of non-haze days ($2.38\pm2.57$ µg m⁻³), accounting for 16.3%
and 22.2% of OC on haze and non-haze days, respectively.
Levoglucosan is widely used as a key tracer for biomass burning emissions (Bhattarai
et al., 2019; Cheng et al., 2013; Xu et al., 2019a). Based on a levoglucosan to OC ratio
of 8.2 % (Zhang et al., 2007a; Fan et al., 2020), the BB-derived OC was $3.40\pm2.09$ µg
m⁻³ and $0.32\pm0.35$ µg m⁻³ during the winter and summer campaigns, respectively. These
results are comparable to BB-derived OC from the CMB in this study. The estimated
BB-derived OC concentration are also comparable with the BB-derived OC during the
same sampling periods in Tianjin (Fan et al., 2020), but higher than those at IAP in
2013-2014 (Kang et al., 2018).. Both of the studies applied the levoglucosan/OC ratio
method to estimate the BB-derived OC although the actual ratio in Beijing air may be
very different to 8.2%. The heavily elevated OC concentration in winter compared to
summer could be a result of increased biomass burning activities for house heating and
cooking in Beijing in addition to the unfavorable dispersion conditions under stagnant
weather conditions in the winter.
In summer, the total OC concentration was highest on 17[th] June. The sudden rise of
OC on this day was attributed to the enhanced biomass burning activities, which led to
the highest level of BB-derived OC and highest BBOC to OC abundance. The
levoglucosan concentration on this day was also the highest in summer, which reached
172 ng m⁻³.

### 3.3.3 Gasoline and diesel vehicles

OC and EC are the key components of traffic emissions (gasoline vehicles & diesel
engines) (Chen et al., 2014; Chuang et al., 2016). Traffic related OC, as represented by
the total sum of OC from gasoline and diesel vehicles, was $2.4\pm2.3$ and $0.39\pm0.22$ µg
m⁻³, and contributed $12.1\pm7.8$% and $6.1\pm3.3$% of OC in winter and summer,
respectively. These results are lower than the contribution of vehicle emissions to OC
(13-20%) in Beijing during 2005 and 2006 (Wang et al., 2009), suggesting traffic
emissions may be a less significant contributor to fine OC in the atmosphere in Beijing
in 2016/2017. By multiplying by OM/OC factors of 2.39 and 1.47 in winter and summer,
respectively, as mentioned in section 2.3, traffic related organic aerosol contributed

8.2±6.5% and 2.3±1.7% of $PM_{2.5}$ in winter and summer, respectively. The summer result was comparable with the vehicular emissions contribution to $PM_{2.5}$ (2.1%) in summer in Beijing, but higher than that in winter (1.5%) in Beijing estimated by using a PMF model (Yu et al., 2019). Gasoline vehicles dominanted the traffic emissions; gasoline vehicle-derived OC was 2.03±1.56 and 0.31±0.16 µg m$^{-3}$ in winter and summer, respectively, which are approximately four times than that in winter (0.54±1.15 µg m$^{-3}$) and summer (0.08±0.16 µg m$^{-3}$) attributed to diesel vehicles. On haze days, gasoline- and diesel-derived OC were 2.35±1.27 and 0.83±1.43 µg m$^{-3}$, respectively, much higher than gasoline- (1.59±1.85 µg m$^{-3}$) and diesel-derived (0.14±0.33 µg m$^{-3}$) OC on non-haze days. Even though diesel vehicles played a less important role in OC emissions, diesel-derived OC on haze days increased by around 6 times above that of non-haze days, and such an increase was much higher than for gasoline, suggesting a potentially important role of diesel emissions on haze formation.

### 3.3.4 Cooking

Cooking is expected to be an important contributor of fine OC in densely populated Beijing, which has a population of over 21 million. The cooking source profile was selected from a study which was carried out in the urban area of another Chinese megacity- Guangzhou, which includes fatty acids, sterols, monosaccharide anhydrides, alkanes and PAHs in particles from the Chinese residential cooking (Zhao et al., 2015). The resultant cooking related OC concentrations were 2.23±2.13 µg m$^{-3}$ and 0.66±0.43 µg m$^{-3}$ in winter and summer, respectively, and both accounted for about 10% to total OC. Cooking OC was 3.23±2.30 µg m$^{-3}$ on winter haze days, around four times higher than that on non-haze days (0.85±0.52 µg m$^{-3}$).

### 3.3.5 Vegetative detritus

Vegetative detritus made a minor contribution to fine particle mass. Its concentration was 0.09±0.08 µg m$^{-3}$ (0.4%) and 0.11±0.08 µg m$^{-3}$ (1.7%) of OC during the winter and summer campaigns, respectively. These contributions are comparable with that in winter (0.5%), but higher than that in summer (0.3%) in urban Beijing during 2006-2007 (Wang et al., 2009). These results are also higher than the plant debris-derived OC in Tianjin in winter 2016 (0.02 µg m$^{-3}$) and summer 2017 (0.01 µg m$^{-3}$), which were calculated based on the relationship of glucose and plant debris and a OM/OC ratio of 1.93 (Fan et al., 2020).

### 3.3.6 Other OC

The Other OC was calculated by subtracting the calculated OC (the sum of OC from seven main sources) from measured OC concentrations. As shown in Table S2, there are four major source categories of OC in Beijing based on the Multi-resolution Emission Inventory for China (MEIC), which include power, industry, residential and transportation (Zheng et al., 2018). In the "industry" category, industrial coal combustion has been resolved by the CMB model. The local emissions of OC from industrial coal in Beijing were zero (shown in Table S2), and hence, the resolved POC from industrial coal combustion in Beijing should be regionally-transported. The MEIC data also show a small industrial oil combustion source. Since the tracers for this are likely to be the same as those for petroleum-derived road traffic emissions in CMB, this

may result in a small overestimation of the latter source. For the industrial processes related OC which have not been resolved by the CMB model, the annual average OC emissions in Beijing were 1161 and 1083 tonnes in 2016 and 2017 respectively, which accounted for 7.7% and 9.0% of the total OC emissions (POC). Therefore, the contribution from industrial processes to the total OC in the atmosphere (POC+SOC) was considered relatively small. The Other OC in this study is likely to be a mixture of predominantly SOC and a small portion of POC from sources such as industrial processes.

The Other OC was 5.3±4.9 and 2.9±1.5 µg m$^{-3}$ in winter and summer, respectively, contributing 24.8% and 43.9% of total measured OC. This is in good agreement with the Other OC estimated by CMB in another study in urban Beijing, for which Other OC contributed 22% and 44% of OC in winter and summer, respectively (Wang et al., 2009). SOC/OC in summer was more than 10% higher than that in summer 2008 in Beijing estimated using a tracer yield method, with the SOC derived from specific VOC precursors (toluene, isoprene, α-pinene and β-caryophyllene) accounting for 32.5% of OC (Guo et al., 2012).

Even though the Other OC concentration was lower in summer, its relative abundance was higher than that in winter, suggesting relatively higher efficiency of SOA formation in summer due to more active photochemical processes under higher temperature and strong radiation. The Other OC on winter haze days was 7.4±5.6 µg m$^{-3}$, approximately 3 times of that on non-haze days (2.5±1.4 µg m$^{-3}$). Other OC is also compared with the SOC estimated by EC-tracer method below.

### 3.3.7 SOC calculated based on the EC-tracer method

EC is a primary pollutant, while OC can originate from both primary sources and form in the atmosphere from gaseous precursors, namely primary organic carbon (POC) and SOC, respectively (Xu et al., 2018). The OC/EC ratios can be used to estimate the primary and secondary carbonaceous aerosol contributions. Usually, OC/EC ratios > 2.0 or 2.2 have been applied to identify and estimate SOA (Liu et al., 2017). In this study, all samples were observed with higher OC/EC ratios (>2.2). SOC in this study was estimated using the equation below, assuming EC comes 100% from primary sources and the OC/EC ratio in primary sources is relatively constant (Turpin and Huntzicker, 1995; Castro et al., 1999):

$$SOC_i = OC_i - EC_i \times (OC/EC)_{pri} \qquad (4)$$

where SOC$_i$, OC$_i$ and EC$_i$ are the ambient concentrations of secondary organic carbon, organic carbon and elemental carbon of sample i, respectively. (OC/EC)$_{pri}$ is the OC/EC ratio in primary aerosols. It is difficult to accurately determining the ratio of (OC/EC)$_{pri}$ for a given area. (OC/EC)$_{pri}$ varies with the contributions of different sources and can also be influenced by meteorological conditions (Dan et al., 2004). In this work, (OC/EC)$_{pri}$ was determined based on the lowest 5% of measured OC/EC ratios for the winter and summer campaigns, respectively (Pio et al., 2011). The average SOC concentrations during summer and winter were calculated and are shown in Table 1. Daily concentrations of Other OC estimated by CMB and SOC estimated by the EC-tracer method in winter and summer are plotted in Fig. 6, as well as their correlation relationship.

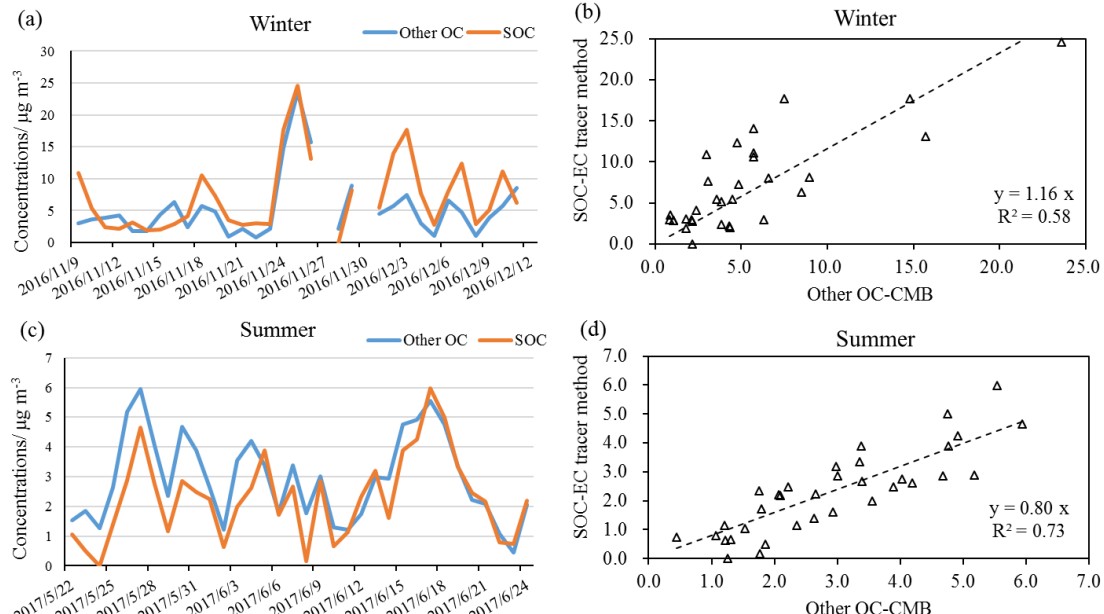

**Figure 6.** Time series of mean values for Other OC estimated by CMB and SOC estimated by the EC-tracer method in winter (a) and summer (c); Correlation relationship between Other OC estimated by CMB and SOC estimated by the EC-tracer method in winter (b) and summer (d).

The average SOC concentrations in winter and summer are presented in Table1. The average SOC concentration during winter was 7.2±5.7 µg m$^{-3}$, accounted for 36.6±15.9% of total OC. The average SOC concentration during summer was one third of that in winter, which was 2.3±1.4 µg m$^{-3}$, accounting for 36.2±16.0% of total OC. The mean SOC concentrations during winter haze and non-haze periods were 10.3±5.7 µg m$^{-3}$ and 2.9±1.4 µg m$^{-3}$, contributing to 34.0±12.0% and 40.5±20.4% of OC during haze and non-haze episodes, respectively. As shown in Fig. 6, the SOC estimated by the EC tracer method followed a similar trend to the Other OC calculated by the CMB model. They were well-correlated in both seasons with $R^2$ of 0.58 and 0.73 in winter and summer samples, respectively and gradients of 1.16 and 0.80. This suggests that the estimates of Other OC calculated from the CMB outputs were reasonable and mainly represented the secondary organic aerosol.

**3.4 Comparison with the source apportionment results in rural Beijing**

The OC source apportionment results in this study are also compared with those in another study conducted at a rural site of Beijing - Pinggu during APHH-Beijing campaigns (Wu et al., 2020). CMB was run based on the results from high-time resolution PM$_{2.5}$ samples that were collected in Pinggu during the same sampling period, but not on identical days. It is valuable to study both rural and urban sites, as both exceed health-based guidelines and require evidence-based mitigation policies which may differ depending on the source apportionment at each. Furthermore, urban air pollution may affect the pollution levels in rural areas (Chen et al., 2020b), and domestic heating and cooking led to high emissions of particles and precursor gases, which may contribute to air pollution in the cities (Liu et al., 2021). The comparison of results is presented in Fig. 7 and Table S3.

As shown in Fig. 7 and Table S3, slightly more OC was explained by CMB at the urban site (75.7%) than the rural site (69.1%) during winter, but less OC was explained at the urban site (56.1%) than the rural site (63.4%) during summer. As at the urban site, biomass burning and coal combustion are important primary sources in rural Beijing. Diesel contributed more to OC at the rural site, while cooking contributed more at the urban site. The rural site also had a larger contribution from vegetative detritus to OC than the urban site. The source contribution estimates from biomass burning at the rural site was approximately 2 and 4 times that at the urban site during winter and summer. In winter, biomass burning contributed a similar percentage of OC at both sites. A higher percentage of OC from biomass burning was found at the rural site than the urban site in summer, possibly because of use of biomass for cooking. For traffic emitted OC, gasoline exceeded diesel at the urban site, while the rural site by contrast has a larger diesel contribution. Industrial CC emitted OC is higher at the urban site during winter, but lower in summer compared to the rural site. The source contribution estimates of residential CC at the urban site is only half that of the rural site in both seasons, and its relative contribution to OC was also lower at the urban site. Coal is widely used for cooking and heating at the villages around the rural site at the time of observations. Cooking accounted for over 10% of OC at the urban site, but less than 5% at the rural site, which is plausible as the urban site is more densely populated.

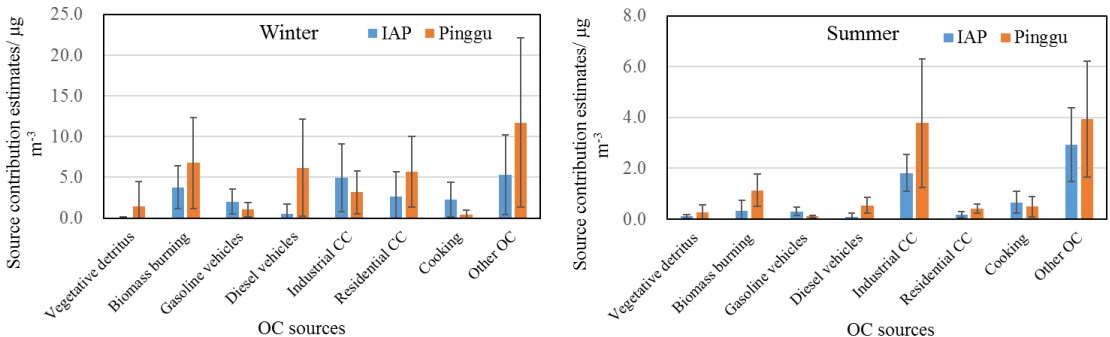

**Figure 7.** Comparison of the source contribution estimates (SCE in µg m$^{-3}$ (%OC)) at IAP with those at a rural site in Beijing- Pinggu

### 3.5 Comparison with source apportionment results from AMS-PMF

Results from AMS-PMF were compared with the CMB source apportionment results to investigate the consistency and potential uncertainties of both methods, and also to provide supplemental source apportionment results (Ulbrich et al., 2009; Elser et al., 2016). Similar comparisons have yielded valuable insights in earlier studies (Aiken et al., 2009; Yin et al., 2015). It is noteworthy that the CMB model was applied to PM$_{2.5}$ samples, while AMS-PMF was applied for NR-PM$_1$ species. This may consequently cause differences in the chemical composition and source attribution between the two methods, as larger particles were not captured by AMS. However, as mentioned in the study of Aiken et al. (2009), the mass concentration between PM$_1$ and PM$_{2.5}$ was small with a reduced fraction of OA and increased fraction of dust. In addition, OC fractions in fine particles were found mostly concentrated in particles <1 µm (Chen et al., 2020a; Zhang et al., 2018; Tian et al., 2020). Hence, the bias was expected to be relatively small. Six factors in non-refractory (NR)-PM$_1$ from the AMS were identified based on the mass spectra measured in winter at IAP by applying a PMF model, including coal combustion OA (CCOA-AMS), cooking OA (COA-AMS), biomass burning OA

(BBOA-AMS) and 3 secondary factors of oxidized primary OA (OPOA-AMS), less-oxidized OA (LOOOA-AMS), and more-oxidized OA (MOOOA-AMS). In summer, the PMF analysis resulted in 5 factors including 2 primary factors of hydrocarbon-like OA (HOA-AMS), cooking OA (COA-AMS) and 3 secondary factors of oxygenated OA (OOA-AMS): OOA1, OOA2, OOA3. These OOA factors were identified by PMF based on diurnal cycles, mass spectra and the correlations between OA factors and other measured species. Three OOA factors showed significantly elevated O/C ratios (0.67-1.48), and correlated well with SIA (R=0.52-0.69). Hence, OOA1, OOA2 and OOA3 represent three types of SOA. Compared to OOA2 and OOA3, OOA1 showed relatively higher f43 (fraction of m/z 43 in OA). In addition, the concentrations of OOA1 and OOA3 were higher in daytime, implying the effect of photochemical processing. The variations of OOA2 tracked well with $C_2H_2O_2^+$ (R=0.89), an aqueous-processing related fragment ion (Sun et al., 2016), indicating that OOA2 was an OA factor associated with aqueous-phase processing. Previous studies suggested that aqueous-phase processing plays an important role in the formation of nitrogen-containing compounds (Xu et al., 2017). The fact that OOA2 with relatively high N/C ratios (0.046) was correlated with several N-containing ions (e.g. $CH_4N^+$, $C_2H_6N^+$, R=0.71-0.77) further supports the above argument. The factor profiles of AMS-PMF in winter and summer are provided in Figs. S5 and S6, respectively.

In order to compare with the source apportionment results of OC in this study from the CMB model, the OA concentrations from the AMS-PMF were converted to OC based on various OA/OC ratios measured in Beijing: 1.35 for CCOA/CCOC (coal combustion organic carbon), 1.31 for HOA/HOC (hydrocarbon-like organic carbon) (Sun et al., 2016), 1.38 for COA/COC (cooking organic carbon), 1.58 for BBOA/BBOC (biomass burning organic carbon) (Xu et al., 2019b), and 1.78 for OOA/OOC (Huang et al., 2010). The concentrations of OA and corresponding OC from AMS-PMF analysis are presented in Table 3. As the AMS data were missing during the period 09[th] - 15[th] November 2016, the comparison of the AMS-PMF and CMB results for this period has been excluded.

**Table 3.** Source contributions of OA and OC ($\mu$g m[-3]) from AMS-PMF results in urban Beijing during winter and summer

| Winter | | | |
|---|---|---|---|
| Factors | Concentrations/$\mu$g m[-3] | Factors | Concentrations/$\mu$g m[-3] |
| CCOA | 6.2±4.4 | CCOC | 4.6±3.3 |
| COA | 5.9±4.1 | COC | 4.3±3.0 |
| BBOA | 6.5±5.8 | BBOC | 4.1±3.7 |
| OPOA | 4.6±2.1 | OPOC | 2.6±1.2 |
| LOOOA | 5.2±5.2 | LOOOC | 2.9±2.9 |
| MOOOA | 8.1±7.0 | MOOOC | 4.6±4.0 |
| OOA[a] | 18.0±13.2 | OOC[d] | 10.1±7.4 |
| OM[b] | 36.7±24.0 | | |
| Summer | | | |
| Factors | Concentrations/$\mu$g m[-3] | Factors | Concentrations/$\mu$g m[-3] |
| HOA | 0.7±0.4 | HOC | 0.5±0.3 |
| COA | 1.8±1.0 | COC | 1.3±0.7 |
| OOA1 | 3.3±1.4 | OOC1 | 1.9±0.8 |

| | | | |
|---|---|---|---|
| OOA2 | 2.4±2.4 | OOC2 | 1.4±1.3 |
| OOA3 | 1.9±1.1 | OOC3 | 1.1±0.6 |
| OOA[c] | 7.6±3.7 | OOC | 4.3±2.1 |
| OM | 10.1±3.9 | | |

[a] *OOA=OPOA+LOOOA+MOOOA;* [b] *OM is organics measured by AMS;* [c] *OOA=OOA1+OOA2+OOA3;*
[d] *OOC=OOC1+OOC2+OOC3*

The CCOA-AMS factor was mainly characterized by m/z of 44, 73 and 115 (Sun et al., 2016). In winter, CCOA-AMS was 6.2±4.4 µg m$^{-3}$, contributing 16.9% of OM. CCOC-AMS was 4.6±3.3 µg m$^{-3}$, which was much lower than the estimated coal combustion OC (7.9±5.2 µg m$^{-3}$, industrial and residential coal combustion OC) by CMB (CCOC-CMB). The time series of CCOC-CMB and CCOC-AMS in Fig. 8 showed a similar trend with relatively good correlation of R$^2$ = 0.71, but coal combustion estimated by CMB was consistently higher than by AMS-PMF, probably because AMS-PMF only resolved the sources of NR-PM$_1$, and some coal combustion particles are larger (Xu et al., 2011). The correlation coefficients (R$^2$) of CCOC-AMS with Cl$^-$ and NR-Cl$^-$ were 0.49 and 0.65, respectively in the winter data.

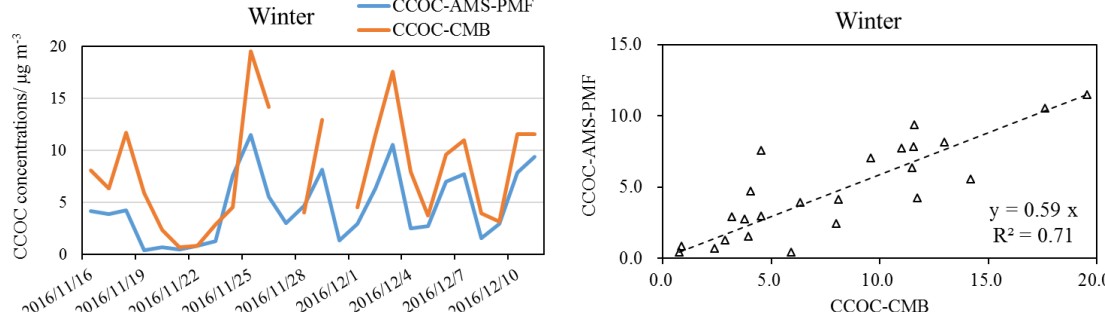

**Figure 8.** Time series and correlation of coal combustion related OC (CCOC) estimated by CMB and CCOC from AMS-PMF analysis

BBOA-AMS in winter was 6.5±5.8 µg m$^{-3}$, contributing 17.7% of OM. This BBOA-AMS factor included a high proportion of m/z 60 and 73, which are typical fragments of anhydrous sugars like levoglucosan (Srivastava et al., 2019). BBOC-AMS was 4.1±3.7 µg m$^{-3}$, which was very close to the estimated BBOC-CMB (3.72±2.79 µg m$^{-3}$, 16.4% of OC) during the same period.

COA-AMS is as a common factor identified in both winter and summer results. It is characterized by high m/z of 55 and 57 in the mass spectrum (Sun et al., 2016). COA-AMS was 5.9±4.1 and 1.8±1.0 µg m$^{-3}$ in winter and summer, respectively, contributing 16.1% and 17.8% of OM. COC-AMS was 4.3±3.0 and 1.3±0.7 µg m$^{-3}$ in winter and summer, respectively, which were almost 2 times of the COC-CMB results for winter (2.20±1.97 µg m$^{-3}$) and summer (0.66±0.43 µg m$^{-3}$). Yin et al. (2015) also reported that COC-AMS was about 2 times of COC-CMB. The overestimation of cooking OC by AMS-PMF could be due to a low relative ionization efficiency (RIE) for cooking OAs (1.4) in AMS while the actual RIE could be higher, such as 1.56-3.06 (Reyes-Villegas et al., 2018), and/or the use of a relatively low OA/OC ratio for cooking (Xu et al., 2021).

HOA-AMS was 0.7±0.4 µg m$^{-3}$ in summer, accounting for 6.9% of OM. HOA-AMS is usually identified based on the high contribution of aliphatic hydrocarbons in this factor, particularly m/z of 27, 41, 55, 57, 69 and 71 (Aiken et al., 2009). This result is lower than that (17% of OM) in rural Beijing during summer 2015 (Hua et al., 2018). HOC-AMS was 0.5±0.3 µg m$^{-3}$ in summer, which is higher than the traffic (gasoline+diesel) emitted OC (0.4±0.2 µg m$^{-3}$) from the CMB model. No obvious correlation was observed between HOC with nitrate and traffic emitted OC from the CMB model during summer.

OOA-AMS concentrations (the sum of all oxidized OA) were 18.0±13.2 and 7.6±3.7 µg m$^{-3}$ in winter and summer, respectively, accounting for 49.0% and 75.2% of OM. The derived OOC-AMS concentrations in winter and summer were 10.1±7.4 and 4.3±2.1 µg m$^{-3}$ in winter and summer, respectively, higher than the Other OC estimated by CMB (Other OC-CMB) in winter (6.1±5.5 µg m$^{-3}$) and summer (2.9±1.5 µg m$^{-3}$) in this study. This could be because AMS-PMF did not resolve HOC in winter and CCOC in summer, which may be mixed with the OOA factors and lead to overestimation of OOC concentrations. The time series and correlation of Other OC-CMB and OOC-AMS is plotted in Fig. 9. A similar temporal trend was found between them, especially in summer, which was also observed with a better correlation (R$^2$=0.73).

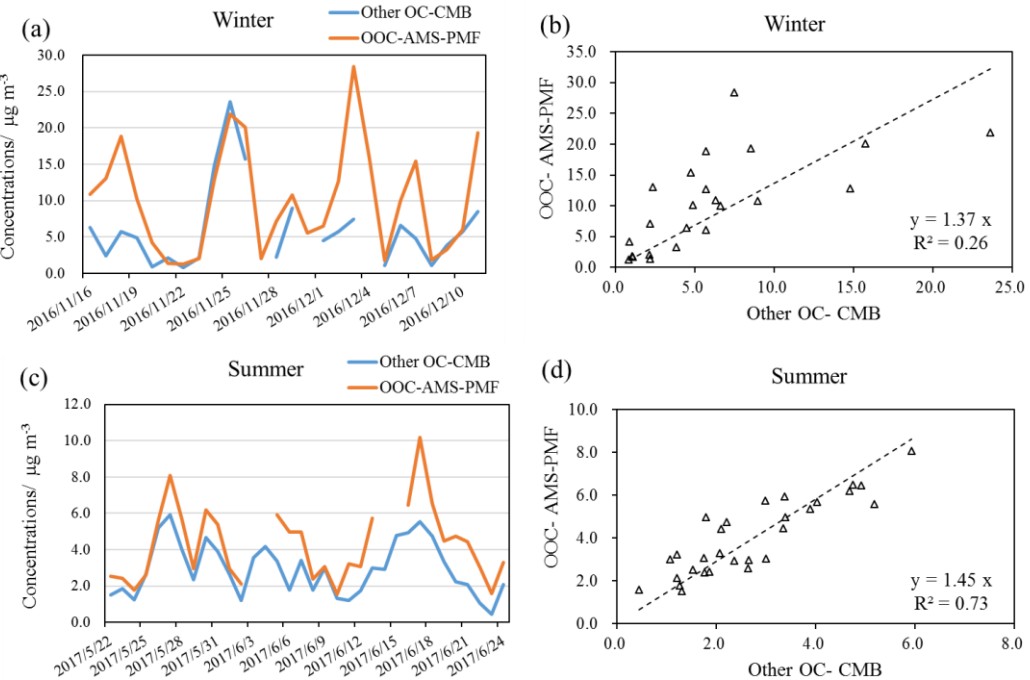

**Figure 9.** Time series of mean values for Other OC estimated by CMB, and OOC estimated by AMS-PMF in winter (a) and summer (c); Correlation relationship between Other OC estimated by CMB and OOC estimated by AMS-PMF in winter (b) and summer (d).

In summary, CMB is able to resolve almost all major known primary OA sources, but AMS-PMF can resolve more secondary OA sources. The AMS-PMF results for major components, such as CCOC-AMS and OOC-AMS agreed well with the results from CMB in the winter. However, discrepancies or poor agreement was found for other sources, such as BBOA-AMS and COA-AMS, although the temporal features were very similar. Furthermore, AMS-PMF did not identify certain sources, probably due to

their relatively small contribution to particle mass. Overall, CMB and AMS-PMF
offered complementary data to resolve both primary and secondary sources.

**3.6 Source contributions to PM$_{2.5}$ from the CMB model**

The source contributions to PM$_{2.5}$ were calculated by multiplication of the fine OC
source estimates from CMB by the ratios of fine OC to PM$_{2.5}$ mass (Table S4), which
were obtained from the same source profiles used for the OC apportionment by CMB
(Zhang et al., 2007b; Wang et al., 2009; Cai et al., 2017; Zhang et al., 2008). For cooking,
vegetative detritus and secondary organic aerosols, OM/OC ratios were applied
considering the low contribution of inorganic species to PM$_{2.5}$ mass from these sources
(Zhao et al., 2007; Bae et al., 2006b). The OM/OC ratios for oxygenated OA were in
the range of 1.85-2.3 (Zhang et al., 2005; Aiken et al., 2008), and the OM/OC ratio was
2.17 in secondary organic aerosols of PM$_{2.5}$ (Bae et al., 2006a). Therefore, an OM/OC
ratio of 2.2 is applied in this study to convert the Other OC to OM. Due to the variability
of the OC/PM$_{2.5}$ ratio in the source profiles, the application using the average OC/ PM$_{2.5}$
ratio of each source to convert the OC to PM$_{2.5}$ in all samples may be subject to
uncertainties, as both organic species and PM$_{2.5}$ mass measurements are subject to
analytical imprecision. Unfortunately, insufficient data are available for a formal
analysis of uncertainty, but errors of around +/- 10% seem very probable. In addition,
instead of OC/PM$_{2.5}$, applying an OM/OC ratio to cooking and vegetative detritus
sources for the calculation may result in an underestimation of PM$_{2.5}$ source
contributions from these sources, because they can also emit inorganic pollutants.
However, cooking emissions are mostly organic and the contribution from vegetative
detritus to PM$_{2.5}$ is very small, so their effects on source contribution estimation here
are considered negligible. The daily PM$_{2.5}$ contribution estimates and seasonal average
source contributions are provided in Fig. S7 and Fig. 10, respectively. Detailed data and
their relative abundance in the reconstructed PM$_{2.5}$ are summarized in Table S5.
As shown in Table S5, PM$_{2.5}$ mass was well explained by those sources which
accounted for 91.9±24.1% and 99.0±19.1% of online PM$_{2.5}$ in winter and summer,
respectively. In the summer, the offline PM$_{2.5}$ is lower than online observations. Thus,
the CMB-based source contributions are more than offline PM$_{2.5}$ mass (121.7±26.6%).
On average, the source contributions in winter ranked as SNA > coal combustion >
Other OM > biomass burning > gasoline & diesel > geological minerals > cooking >
vegetative detritus; in summer these ranked as SNA > other OM > coal combustion >
geological minerals > cooking > gasoline & diesel > biomass burning > vegetative
detritus.
Zheng et al. (2005) investigated the seasonal trends of PM$_{2.5}$ source contributions in
Beijing during 2000 applying a CMB model. In winter (January), the contributions from
coal combustion, biomass burning, diesel & gasoline, vegetative detritus to PM$_{2.5}$ were
9.55 µg m$^{-3}$ (16% of PM$_{2.5}$ and hereafter), 5.8 µg m$^{-3}$ (9%), 3.85 µg m$^{-3}$, 0.33 µg m$^{-3}$,
respectively. Contributions from gasoline, diesel, coal combustion and biomass burning
were enhanced in Beijing during winter in 2016 compared to 2000, while the
contribution from vegetative detritus basically remained similar. In summer (July) 2000,
coal combustion contributed 2% of PM$_{2.5}$ (2.39 µg m$^{-3}$), much less than that in summer
2016 of this study. The contribution from diesel & gasoline (7.78 µg m$^{-3}$, Zheng et al.,
2005) was approximately 10 times of that in 2016 (0.8 µg m$^{-3}$). Similarly, contributions
from vegetative detritus and biomass burning were small and insignificant.

Zhou et al. (2017) estimated that coal combustion contributions in winter and summer of Beijing-Tianjin-Hebei area in 2013 were 15.9 μg m$^{-3}$ and 2.1 μg m$^{-3}$, respectively, which are comparable with those in this study. These results are also comparable with the PMF-resolved coal and oil combustion in Beijing during winter (17.4 μg m$^{-3}$) and summer (2.2 μg m$^{-3}$) in 2010 (Yu et al., 2013). SNA contributed 52.7 and 26.4 μg m$^{-3}$ of PM$_{2.5}$ during winter (January) and summer (July), respectively (Yu et al., 2013), which are much higher than those in this study. It is noteworthy that a severe haze pollution event occurred during January 2013, which was characterized by high concentrations of sulfate and nitrate in several studies (Zhou et al., 2017; Han et al., 2016). The contribution from biomass burning in winter is consistent (8.5 μg m$^{-3}$) with this study (8.9 μg m$^{-3}$), but higher in summer (2.6 μg m$^{-3}$) (0.8 μg m$^{-3}$). The cooking source contributed 4.8 and 1.3 μg m$^{-3}$ in PM$_{2.5}$ during winter and summer 2013, respectively, which is also comparable with this study.

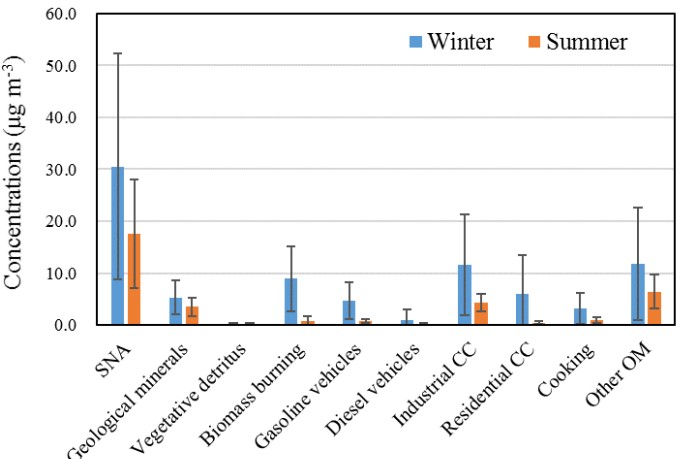

**Figure 10.** Seasonal average PM$_{2.5}$ source contribution estimates from the CMB model

## 4 Conclusions

Carbonaceous aerosols contributed approximately 59% and 41% of reconstructed PM$_{2.5}$ in winter and summer at the urban IAP site in Beijing. The OC and EC concentrations were comparable with more recent studies (Fan et al., 2020; Qi et al., 2018), but lower than those before 2013 (Yang et al., 2016; Dan et al., 2004), suggesting the effectiveness of air pollution control measures since 2013 (Vu et al., 2019; Zhang et al., 2019). CMB modelling showed that in the winter 2016, the top three primary contributors to PM$_{2.5}$-OC were coal combustion (35%), biomass burning (17%), and traffic (12%); these were in the same order with that at the rural site during the same study period: coal combustion (29%), biomass burning (18%), and traffic (17%) (Wu et al., 2020). In the summer 2017, the top three primary contributors to PM$_{2.5}$-OC were coal combustion (32%), cooking (11%), and traffic (6%); these were different to that at the rural site during the same study period: coal combustion (38%), biomass burning (11%), and traffic (7%) (Wu et al., 2020). The Other OC, which was well-correlated (R$^2$: 0.6~0.7; slope: 0.8~1.2) with the secondary OC (SOC) estimated based on the EC-tracer method, accounted for 25% and 44% of OC at urban site and 31% and 37% of

OC at rural site during winter and summer, respectively. Although the annual average
$PM_{2.5}$ levels in Beijing reduced from 88 µg m$^{-3}$ in year 2013 to 58 µg m$^{-3}$ in year 2017
(Vu et al., 2019), and the deweathered concentration of $PM_1$ decreased by -38% in 2017
comparing to 2007 (Zhang et al., 2020), our CMB modelling results indicate that the
coal combustion and biomass burning still remained the dominant primary OC sources
in winter 2016 and summer2017, with road traffic ranked as the third highest. Cooking
was a more significant source of OC than biomass burning at the urban site during
summer. Compared to other CMB studies in Beijing, our study revealed an increase of
the contributions from coal combustion, biomass burning and traffic to $PM_{2.5}$ in winter
2016 compared to winter 2000, while those in this study remained similar compared to
winter 2013. Sulfate, nitrate and ammonium concentrations were significantly lower in
this study compared to 2013 (Zheng et al., 2005; Zhou et al., 2017). It is however
notable that there is a broad consistency in the findings of the CMB studies, whereas
the more numerous studies which have used PMF come to rather diverse conclusions
(Srivastava et al., 2020).

*Data availability.* The data in this article are available from the corresponding authors
upon request.

*Author contributions.* JX did the CMB modelling and drafted the paper with the help
of ZS, RMH and all co-authors. DL, TVV conducted the laboratory analysis of organics
and inorganics, respectively. XW, YZ provided the CMB source profiles. YS provided
the AMS-PMF data.

*Competing interests.* The authors have no conflict of interests.

*Acknowledgement.* This research was funded by the UK Natural Environment Research
Council (NERC, NE/N007190/1; NE/R005281/1) and Royal Society Advanced
Fellowship (grant no: NAF\R1\191220).

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
