# Peer review of "Source Apportionment of Fine Organic Carbon at an Urban Site of"

_Atmospheric Chemistry and Physics, 2020_

## Referee Comment (RC1) · Anonymous Referee #1 · 27 Dec 2020

GENERAL COMMENTS The manuscript presents a study conducted in 2016-2017 in Beijing to estimate the sources of OC in the fine PM fraction using the CMB source apportionment technique. The analytical procedures are described in detail and the manuscript presents a great deal of results which are compared with previous studies in the same or nearby areas. The study focuses mainly to the fine OC fraction. This should be better reflected in the title. In addition, the objectives of the study should be better specified. The apportionment of PM2.5 is a secondary outcome of the study based on OM/PM2.5 ratios from the literature, which introduces a considerable amount of uncertainty. This aspect of the methodology should be clarified to the reader in the presentation of the results. In addition, the reliability of the CMB in relation to other

source apportionment techniques should be better documented. Despite the number of samples (31 in winter and 34 in summer) is coherent with the CMB technique, it is quite limited to represent the variability of meteorological and emission situations in a complex situation like a megacity. Some of the presented results need more discussion to explain what is their relevance for the purposes of the study. For instance: What is the added value of comparing Beijing with a rural location? What is the importance of distinguishing between haze and non-haze days? The readability of the text could be improved by removing the repetition of data already presented in figures and tables. The conclusions should explain what is the added value of this study with respect to the previous knowledge on PM2.5/OC sources in Beijing. In addition, the authors should underline how the combination of two different approaches (CMB and PMF) contribute to obtain more complete/robust estimations.

SPECIFIC COMMENTS Abstract: the abbreviation SNA is not explained Abstract Line 46: this sentence is misleading as the CMB did not apportion any source of secondary OA. Line 73: This statement is not fully correct. At present, there are many numerical tests to support the decisions about the best solution in PMF. The citation here is out of date for this topic. Line 78: "composed" instead of "comprised". Line 80: Explain where. Line 86: Provide references about the reliability of CMB compared to other source apportionment methods. Line 99 and foll.: The list of source profiles is redundant here as it is mentioned in the methodological section. Line 111: Did you check if the number of samples is enough to describe the average conditions in the studied area? Line 137: Federal Reference Method of which country? Please, provide a reference. Line 142: EUSAAR2 method, a reference is needed. Line 215: How did you check that no relevant sources are missing? Line 215: The used source profiles should be provided as table (e.g. suppl. info). Did you check that the sources do not vary considerably within the sampling interval (e.g. winter, summer). Line 256 -257: The period does not match the one described in section 2.5 to compare the results of the two studies you have to align the two time windows. Line 260: What's the importance of distinguishing haze and non-haze and how was the threshold between the two defined? Line 280:

Correlation could be also caused by dispersion- dilution mechanisms. Line 303: This statement is not supported by the data. The agreement is only good in winter for offline and in summer for online. Line 312 and foll.: This behavior could also be explained by the loss of semivolatiles from PTFE filters Line 326 and foll.: Section 3.3 seems not to be in line with the objective of the study. In the introduction you claim that the study focuses on sources of PM2.5 with particular reference to SOA. However, the focus of the study is on OC sources. Line 357: Coal cannot be considered a single contributor but as a single fuel type. Line 370: Cl- has many sources, not a good candidate for tracer. Line 372: In Fig 5 there is no clear relationship between Cl- and coal combustion in summer. Line 529: Provide more information about this site. Why is it important for this study? Table 3: It is too complicated to read, I suggest to convert it into a bar plot with error bars Line 546: More discussion is needed about the different size fractions used in the two methods and how this affects the single sources. Line 564: What is the meaning of OOA1,2 and 3? The profiles of the AMS sources should be shown in a figure Table 4: Please, add the names to the single columns. Line 631 and foll.: This calculation of PM2.5 contributions using OC/PM2.5 ratios from the literature introduces a considerable amount of uncertainty because of the variability of such factors. The uncertainty of these results should be clearly illustrated to the reader. Lines 633-638: It is not necessary to repeat the information that is already available in Table S3. Line 649. Specify what does the range after the value means. Is it a confidence interval, a standard deviation or something else? Does it includes the uncertainty of the OC/PM2.5 ratios? Line 652 and foll.: Since these results are in a table it is not necessary to list them here. Figure 9: I suggest to replace the daily data with the overall averages (now in table S4). Conclusions: this section is mandatory according to the instructions for authors of this journal

---

## Referee Comment (RC2) · Anonymous Referee #2 · 1 Jan 2021

Review of ACP-2020-1020

In the manuscript by Xu et al., the atmospheric fine particle samples from winter and summer at an urban site in Beijing were collected and analyzed. The source apportionment analysis from the Chemical mass balance (CMB) model shows comparable results with that of Positive Matrix Factorization (PMF) analysis of co-located Aerosol Mass Spectrometer (AMS). The paper provides useful scientific evidence to the source contributions in Beijing. However, there are still a few important issues that need to be clarified.

Major issues: One of the major issues in the paper is that the AMS analyzed the

non-refractory fraction of PM1, but the CMB method was applied to PM2.5 samples. The difference in particle size segments is an important factor that could contribute significantly to the difference between the results from these two source apportionment methods. However, the authors didn't give any discussion about it. Especially, as shown in Figure 8, for the AMS-PMF derived OOA concentrations (in PM1), which generally show slightly higher concentrations than other OC estimated by CMB methods (in PM2.5). The authors have to give some reasonable explanations for this.

P10, line 307-313: It is biased to attribute all deviations to the uncertainty of experimental measurements for heavily polluted periods or the sampling artifacts. Are the reconstructed PM2.5 concentrations absolutely right?

Minor suggestions: Page 3, line63-64:"A better understanding of the sources of PM2.5 in Beijing is essential to provide scientific evidence to control the PM2.5 pollution." Such an expression is inaccurate.

Page 3, the last paragraph: Please pay attention to the logical order of these sentences. For example, you have to introduce the CMB method was used to do the source apportionment first, then state how the source profiles were determined. Similar mistakes were also found in P5, line 143-146.

P5, line 159-160: Please describe exactly how much is "A portion of the filters." So did the other analyses, such as inorganic components using Ion Chromatograph and ICP-MS in Section 2.2.3 (P6, second paragraph).

P7, First paragraph: Please rephrase and reorganized these sentences.

P7, line 241-244: Rewording, please!

P18, line 560-565, and P7 line 245-251: Unnecessary to repeat this.

P19-P20: when describing the comparisons between the results from CMB and AMS-PMF, the language and symbols sometimes are quite confusing; please reorganize them.

---

## Author Comment (AC1) · 10 Mar 2021

**Point-by-Point Response to Reviewers' Comments**

Manuscript Ref: acp-2020-1020

Title: Source Apportionment of Fine Aerosol at an Urban Site of Beijing using a Chemical Mass Balance Model

Journal: Atmospheric Chemistry and Physics

**Comments from Reviewer #1**

**General Comments:**

The manuscript presents a study conducted in 2016-2017 in Beijing to estimate the sources of OC in the fine PM fraction using the CMB source apportionment technique. The analytical procedures are described in detail and the manuscript presents a great deal of results which are compared with previous studies in the same or nearby areas. The study focuses mainly to the fine OC fraction. This should be better reflected in the title. In addition, the objectives of the study should be better specified. The apportionment of $PM_{2.5}$ is a secondary outcome of the study based on $OM/PM_{2.5}$ ratios from the literature, which introduces a considerable amount of uncertainty. This aspect of the methodology should be clarified to the reader in the presentation of the results. In addition, the reliability of the CMB in relation to other source apportionment techniques should be better documented. Despite the number of samples (31 in winter and 34 in summer) is coherent with the CMB technique, it is quite limited to represent the variability of meteorological and emission situations in a complex situation like a megacity. Some of the presented results need more discussion to explain what is their relevance for the purposes of the study. For instance: What is the added value of comparing Beijing with a rural location? What is the importance of distinguishing between haze and non-haze days? The readability of the text could be improved by removing the repetition of data already presented in figures and tables. The conclusions should explain what is the added value of this study with respect to the previous knowledge on $PM_{2.5}/OC$ sources in Beijing. In addition, the authors should underline how the combination of two different approaches (CMB and PMF) contribute to obtain more complete/robust estimations.

**General response:** We thank the reviewer for very constructive suggestions and are pleased to respond.

(1) This study focuses mainly to the fine OC fraction. As suggested by the reviewer, to better reflect that in the title, the original title "Source Apportionment of Fine Aerosol at an Urban Site of Beijing using a Chemical Mass Balance Model" has been changed to "Source Apportionment of Fine Organic Carbon at an Urban Site of Beijing using a Chemical Mass Balance Model"

(2) The objectives of this study are specified and added in the revised manuscript, please see **lines 111-115** in the revised manuscript, which are also copied below:

"The objectives of this study are: 1) to quantify the contributions of pollution sources to OC by applying a CMB model and compare them with those at a rural site of Beijing; 2) to compare the source apportionment results by CMB with those from Aerosol Mass Spectrometer-PMF analysis (AMS-PMF), to improve our understanding of different sources of OC."

(3) We agree with the reviewer that the calculation of $PM_{2.5}$ contributions using $OC/PM_{2.5}$ ratios from the applied source profiles introduces uncertainty. For example, due to the variability of the OC/ $PM_{2.5}$ ratio in the source profiles, the application using the average OC/ $PM_{2.5}$ ratio of each source to convert the OC to $PM_{2.5}$ in all samples may be subject to uncertainties, as both organic species and $PM_{2.5}$ mass measurements are subject to analytical imprecision. Unfortunately, insufficient data are available for a formal analysis of uncertainty, but errors of around +/- 10% seem very probable. In addition, instead of $OC/PM_{2.5}$, applying an OM/OC ratio to cooking and vegetative detritus sources for the calculation may result in an underestimation of $PM_{2.5}$ source contributions from these sources, because they can also emit inorganic pollutants. However, cooking emissions are mostly organic and the contribution from vegetative detritus to $PM_{2.5}$ is very small, so their effects on source contribution estimation here are considered negligible. Please also see response to Comment 26.

(4) The repetition of data that are already presented in figures and tables has been removed to improve the readability of the text. Please see the responses to specific comments below.

(5) It is valuable to separate haze and non-haze days, as if policy is directed at reducing the number of haze days, it is essential to know the contributing sources, which may differ from non-haze days. There is also value in studying both rural and urban sites, as both exceed health-based guidelines and require evidence-based mitigation policies which may differ depending on the source apportionment at each. Furthermore, urban air pollution may affect the pollution levels in rural areas (Chen et al., 2020), and domestic heating and cooking led to high emissions of particles and precursor gases, which may contribute to air pollution in the cities (Liu et al., 2021). Please also see responses to Comment 14 and 21.

(6) The added value of this study with respect to the previous knowledge on $PM_{2.5}$/OC sources in Beijing is provided in the conclusion and copied below:

"Although the annual average $PM_{2.5}$ levels in Beijing reduced from 88 $\mu g\ m^{-3}$ in year 2013 to 58 $\mu g\ m^{-3}$ in year 2017 (Vu et al., 2019), and the deweathered concentration of $PM_1$ decreased by -38% in 2017 comparing to 2007 (Zhang et al., 2020), our CMB modelling results indicate that the coal combustion and biomass burning still remained the dominant primary OC sources in winter 2016 and summer2017, with road traffic ranked as the third highest. Cooking was a more

significant source of OC than biomass burning at the urban site during summer. Compared to other CMB studies in Beijing, our study revealed an increase of the contributions from coal combustion, biomass burning and traffic to $PM_{2.5}$ in winter 2016 compared to winter 2000, while those in this study remained similar compared to winter 2013."

(7) Receptor modelling techniques are still an active area of research. None is infallible, and although they are rarely compared, they may not give consistent findings when they are. Such comparisons are important as they highlight the areas of uncertainty. In this study, most factor contributions were in good agreement, but some of the factor contributions from CMB and AMS-PMF varied significantly; for example, cooking OC was 2 times different between the two methods, and it is important to highlight such uncertainties.

(8) Variability of meteorological and emission situations - see response to comment 8. We recognize that our study did not cover all seasons, but the winter and summer study periods represent the typical emission situations in respective seasons.

**Specific Comments:**

Comment 1 - Abstract: the abbreviation SNA is not explained

**Response:** In abstract line 29-30, the original phrase "Secondary inorganic ions (sulfate, nitrate, ammonium; SNA)" has been changed to "Secondary inorganic ions (sulfate, nitrate, ammonium- SNA)" to avoid confusion. Please see **line 30** in the revised manuscript.

Comment 2 - Abstract Line 46: this sentence is misleading as the CMB did not apportion any source of secondary OA.

**Response:** The original sentence "The CMB was found to resolve more primary OA sources than AMS-PMF but the latter apportioned more secondary OA sources." has been modified as "The CMB was found to resolve more primary OA sources than AMS-PMF but the latter could apportion secondary OA sources." to avoid any misunderstanding. Please see **lines 45-46** in the revised manuscript.

Comment 3 - Line 73: This statement is not fully correct. At present, there are many numerical tests to support the decisions about the best solution in PMF. The citation here is out of date for this topic.

**Response:** The original sentence has been modified, and new citations have been added.

The original sentence "This approach has some underlying issues: firstly, PMF requires a relatively large sample size; and a "best" solution of achieved factors is subjective (Ulbrich et al., 2009);" has been revised as "This approach has some underlying challenges. For example, PMF requires a relatively large sample size and a "best" solution of achieved factors requires critical assessment of its mathematical parameters

and evaluation of the physical reasonability of the factor profiles (de Miranda et al., 2018; Ikemori et al., 2021; Oduber et al., 2021);" Please see **lines 73-77** in the revised manuscript.

Added references:

de Miranda, R. M., de Fatima Andrade, M., Dutra Ribeiro, F. N., Mendonça Francisco, K. J., and Pérez-Martínez, P. J.: Source apportionment of fine particulate matter by positive matrix factorization in the metropolitan area of São Paulo, Brazil, Journal of Cleaner Production, 202, 253-263, https://doi.org/10.1016/j.jclepro.2018.08.100, 2018.

Ikemori, F., Uranishi, K., Asakawa, D., Nakatsubo, R., Makino, M., Kido, M., Mitamura, N., Asano, K., Nonaka, S., Nishimura, R., and Sugata, S.: Source apportionment in PM2.5 in central Japan using positive matrix factorization focusing on small-scale local biomass burning, Atmospheric Pollution Research, https://doi.org/10.1016/j.apr.2021.01.006, 2021.

Oduber, F., Calvo, A. I., Castro, A., Blanco-Alegre, C., Alves, C., Calzolai, G., Nava, S., Lucarelli, F., Nunes, T., Barata, J., and Fraile, R.: Characterization of aerosol sources in León (Spain) using Positive Matrix Factorization and weather types, Science of The Total Environment, 754, 142045, https://doi.org/10.1016/j.scitotenv.2020.142045, 2021.

Comment 4 - Line 78: "composed" instead of "comprised".

**Response:** This is revised. Please see **line 80** in the revised manuscript.

Comment 5 - Line 80: Explain where.

**Response:** The original sentence "OM was the largest contributor to PM$_{2.5}$ mass, which accounted for 30%-60% of PM$_{2.5}$ (Song et al., 2007; He et al., 2001; Huang et al., 2014), and can contribute up to 90% of submicron PM mass (Zhou et al., 2018)." has been modified to "OM was the largest contributor to PM$_{2.5}$ mass, which was reported to account for 30%-50% of PM$_{2.5}$ in some Chinese cities such as Beijing, Guangzhou, Xi'an and Shanghai (Song et al., 2007; He et al., 2001; Huang et al., 2014), and can contribute up to 90% of submicron PM mass in Beijing (Zhou et al., 2018)." Please see **lines 83-86** in the revised manuscript.

Comment 6 - Line 86: Provide references about the reliability of CMB compared to other source apportionment methods.

**Response:** Regarding the reliability of the CMB, more discussion has been included and references have been added.

The original sentence "A few studies have also applied a Chemical Mass balance (CMB) model for source apportionment of PM in Beijing." has been modified to "Chemical Mass balance (CMB) model has been used for source apportionment of PM worldwide, including in the US (Antony Chen et al., 2010), UK (Yin et al., 2015), and China (Chen et al., 2015b). The CMB model assumes that source profiles remain unchanged between the emitter and receptor (Sarnat et al., 2008; Viana et al., 2008). Xu et al. (2021)

compared the source apportionment results of fine particles by multiple receptor modelling approaches, and found that CMB can provide the most complete and representative source apportionment of Beijing aerosols. A few studies have applied a CMB model for source apportionment of PM in Beijing (Zheng et al., 2005; Liu et al., 2016; Guo et al., 2013; Wang et al., 2009)." Please see **lines 90-98** in the revised manuscript.

Comment 11 - Line 215: How did you check that no relevant sources are missing?

**Response:** The missing sources have been checked and discussed in the manuscript.

Firstly, as mentioned in section 3.3.6 "Other OC", there are four major source categories of OC in Beijing based on the Multi-resolution Emission Inventory for China (MEIC), which include power, industry, residential and transportation. In our study, the OC sources applied in the CMB model were biomass (straw and wood) burning, gasoline and diesel vehicles, industrial and residential coal combustion, cooking and vegetative detritus, which included the four major source categories of OC in Beijing as described above.

Secondly, we also mentioned in section 3.3.6 that industrial coal combustion has been resolved by the CMB model, but there are industrial processes related emissions of OC which have not been resolved by the CMB model. The annual average OC emissions

from these sources in Beijing were 1161 and 1083 tonnes in 2016 and 2017 respectively, which accounted for 7.7% and 9.0% of the total OC emissions (POC). Therefore, the contribution from industrial processes to the total OC in the atmosphere (POC+SOC) was considered relatively small.

Thirdly, because the CMB model explained the majority of fine OC in winter (75.7%) and summer (56.1%). In addition, the Other (unexplained) OC was found correlated with the secondary OC (SOC) estimated by the EC-tracer method, with correlation coefficients ($R^2$) of 0.58 and 0.73, and slopes of 1.16 and 0.80 in winter and summer, respectively. Hence, we believe the Other OC in this study is likely to be a mixture of predominantly SOC and a small portion of POC from missing sources such as industrial processes.

Comment 12 - Line 215: The used source profiles should be provided as table (e.g. suppl. info). Did you check that the sources do not vary considerably within the sampling interval (e.g. winter, summer).

**Response:** We didn't provide the source profiles in this study because the same source profiles have already been provided in Table S1 of Wu et al. (2020). To be more informative as suggested by the reviewer, we have added the following sentence in section 2.4:

"The source profiles with EC and organic tracers used in the CMB model were provided in Table S1 of Wu et al. (2020)." Please see **lines 234-235** in the revised manuscript.

Regarding the sources in Beijing during the sampling interval, as mentioned above, we have included the major sources of aerosols in Beijing, the CMB model explained the majority of fine OC in winter (75.7%) and summer (56.1%). Zheng et al. (2005) studied the seasonal trends in $PM_{2.5}$ source contributions in Beijing, and the major sources for organics in different seasons were the same as in this study. If the source contributions varied significantly between winter and summer, this would be reflected in the source apportionment results. If the sources types varied significantly, then the missing sources will cause less explained OC, and the Other OC would be poorly correlated with SOC by other methods. Hence, regarding source types changing within a season, we believe that we have included all relevant major source types in our CMB model. Source contributions changed significantly from day-to-day, largely due to variations in meteorology, as seen in the outputs of the CMB, but our study focuses on seasonal campaign averages. The representativeness of the meteorological conditions during the campaigns are given in Shi et al., (2019).

Comment 20 - Line 370: Cl⁻ has many sources, not a good candidate for tracer. Line 372: In Fig 5 there is no clear relationship between Cl⁻ and coal combustion in summer.

**Response:** We agree with the reviewer that Cl⁻ has many sources such as sea-salt, coal combustion and biomass burning, and is not a good candidate as a tracer for coal combustion. The reasons for no clear relationship between Cl⁻ and coal combustion in summer are discussed in the revised manuscript. Please see details below.

Beijing is an inland city, so the contribution of marine aerosols to particulate Cl⁻ is considered minor. This is also supported by the higher $Cl^-/Na^+$ mass ratios in winter (10.1±4.8) and summer (2.7±1.8) than sea water (1.81), indicative of significant contributions from anthropogenic sources (Bondy et al., 2017). Yang et al. (2018) also reported that the contribution of sea-salt aerosol to fine particulate chloride was negligible in inland areas even during summer. Hence, particulate Cl⁻ in this study was mainly from anthropogenic sources. The moderately good correlation of Cl⁻ and coal combustion OC in winter could be due to enhanced coal combustion activities in this season. No correlation of Cl⁻ and coal combustion OC in summer was a result of less coal combustion, and the abundance of Cl⁻ in summer was more influenced by other sources. Therefore, we have modified the discussion accordingly.

The original discussion:

"Chloride has been considered as a tracer for coal combustion (Chen et al., 2014). The time series of OC from coal combustion (OC-CC) and Cl⁻ during winter and summer of Beijing are shown in Fig. 5. OC-CC and Cl⁻ exihibited similar trends in both seasons. The correlation coefficient ($R^2$) between OC-CC and Cl⁻ during winter was 0.62 but there is no significant correlation between the two during the summer campaign. This is probably related to the semi-volatility of ammonium chloride, which is liable to evaporate in summer (Pio and Harrison, 1987). A similar phenomenon has been observed in Delhi (Pant et al., 2015). "

The revised discussions:

"Coal combustion is also a major source for particulate chloride (Chen et al., 2014). Because Beijing is an inland city, the contribution of marine aerosols to particulate Cl⁻ is considered minor, which is also supported by the higher $Cl^-/Na^+$ mass ratios in winter (10.1±4.8) and summer (2.7±1.8) than sea water (1.81), indicative of significant contributions from anthropogenic sources (Bondy et al., 2017). Yang et al. (2018) also reported that the contribution of sea-salt aerosol to fine particulate chloride was negligible in China inland areas even during summer. Hence, Cl⁻ in this study was mainly from anthropogenic sources. The time series of OC from coal combustion (OC-CC) and Cl⁻ during winter and summer of Beijing are shown in Fig. 5. OC-CC and Cl⁻ exhibited similar trends in both seasons. The correlation coefficient ($R^2$) between OC-CC and Cl⁻ during winter was 0.62, which could be attributed to enhanced coal combustion activities in this season. No significant correlation between the two was found during the summer campaign, indicating the abundance of Cl⁻ in summer was more influenced by other sources, probably including biomass burning. In addition, due to the semi-volatility of ammonium chloride, it is liable to evaporate in summer (Pio

and Harrison, 1987). A similar phenomenon has been observed in Delhi (Pant et al., 2015). "

Please see **lines 391-407** in the revised manuscript.

Added reference:

Bondy, A. L., Wang, B., Laskin, A., Craig, R. L., Nhliziyo, M. V., Bertman, S. B., Pratt, K. A., Shepson, P. B., and Ault, A. P.: Inland Sea Spray Aerosol Transport and Incomplete Chloride Depletion: Varying Degrees of Reactive Processing Observed during SOAS, Environ. Sci. Technol., 51, 9533-9542, 10.1021/acs.est.7b02085, 2017.

Yang, X., Wang, T., Xia, M., Gao, X., Li, Q., Zhang, N., Gao, Y., Lee, S., Wang, X., Xue, L., Yang, L., and Wang, W.: Abundance and origin of fine particulate chloride in continental China, Science of The Total Environment, 624, 1041-1051, https://doi.org/10.1016/j.scitotenv.2017.12.205, 2018.

Comment 21 - Line 529: Provide more information about this site. Why is it important for this study?

**Response:**

More information about this site has been provided in section 2.1 "Aerosol sampling". The original discussion "The location of a rural site in Beijing - Pinggu during the APHH-China campaigns is also shown in Fig. 1. Other information regarding the sampling site is described elsewhere (Shi et al., 2019)." has been modified as "The location of a rural site in Beijing - Pinggu during the APHH-China campaigns is also shown in Fig. 1. The rural site in Xibaidian village in Pinggu is about 60 km away from IAP and 4 km north-west of the Pinggu town centre. It is surrounded by trees and farmland with several similar small villages nearby. A provincial highway is approximately 500 m away on its eastside running north-south. This site is far from industrial sources and located in a residential area. Other information regarding the sampling site is described elsewhere (Shi et al., 2019)." Please see **lines 125-131** in the revised manuscript.

Both IAP and Pinggu sites were chosen in the APHH-Beijing programme for the study of emission sources, atmospheric processes and health effects of air pollution in Beijing.

It is valuable to study both rural and urban sites, as both exceed health-based guidelines and require evidence-based mitigation policies which may differ depending on the source apportionment at each. Furthermore, urban air pollution may affect the pollution levels in rural areas (Chen et al., 2020), and domestic heating and cooking led to high emissions of particles and precursor gases, which may contribute to air pollution in the cities (Liu et al., 2021). Hence, to address the importance of including the rural site, the original discussion in section 3.4 "The OC source apportionment results in this study are also compared with those in another study conducted at a rural site of Beijing - Pinggu during APHH-Beijing campaigns (Wu et al., 2020). CMB was run based on the results from high-time resolution $PM_{2.5}$ samples that were collected in Pinggu during the same sampling period, but not on identical days. The comparison results are presented in Table 3." has been revised as

"The OC source apportionment results in this study are also compared with those in another study conducted at a rural site of Beijing - Pinggu during APHH-Beijing campaigns (Wu et al., 2020). CMB was run based on the results from high-time resolution $PM_{2.5}$ samples that were collected in Pinggu during the same sampling period, but not on identical days. It is valuable to study both rural and urban sites, as both exceed health-based guidelines and require evidence-based mitigation policies which may differ depending on the source apportionment at each. Furthermore, urban air pollution may affect the pollution levels in rural areas (Chen et al., 2020), and domestic heating and cooking led to high emissions of particles and precursor gases, which may contribute to air pollution in the cities (Liu et al., 2021). The comparison of results is presented in Fig. 7 and Table S3." Please see **lines 558-568** in the revised manuscript.

Comment 25 - Table 4: Please, add the names to the single columns.

**Response:** The names of the single columns have been added in the revised Table 4.

Comment 26 - Line 631 and foll.: This calculation of PM2.5 contributions using OC/PM2.5 ratios from the literature introduces a considerable amount of uncertainty because of the variability of such factors. The uncertainty of these results should be clearly illustrated to the reader.

**Response:** We agree with the reviewer that the calculation of PM2.5 contributions using OC/PM2.5 ratios from the applied source profiles introduces a considerable amount of uncertainty. Hence, apart from the discussion in the original manuscript, we also added more discussion regarding the uncertainties of the results.

Original discussion:

"Instead of OC/PM$_{2.5}$, applying an OM/OC ratio for the calculation may result in an underestimation of PM$_{2.5}$ source contributions, because sources like cooking and vegetative detritus can also emit inorganic pollutants. However, cooking emissions are mostly organic and the contribution from vegetative detritus to PM$_{2.5}$ is very small, their effects on source contribution estimation here are considered negligible."

Modified discussion:

"Due to the variability of the OC/ PM$_{2.5}$ ratio in the source profiles, the application using the average OC/ PM$_{2.5}$ ratio of each source to convert the OC to PM$_{2.5}$ in all samples may be subject to uncertainties, as both organic species and PM$_{2.5}$ mass measurements are subject to analytical imprecision. Unfortunately, insufficient data are available for a formal analysis of uncertainty, but errors of around +/- 10% seem very probable. In addition, instead of OC/PM$_{2.5}$, applying an OM/OC ratio to cooking and vegetative detritus sources for the calculation may result in an underestimation of PM$_{2.5}$ source contributions from these sources, because they can also emit inorganic pollutants. However, cooking emissions are mostly organic and the contribution from vegetative detritus to PM$_{2.5}$ is very small, so their effects on source contribution estimation here are considered negligible."

Please see **lines 711-722** in the revised manuscript.

Comment 27 - Lines 633-638: It is not necessary to repeat the information that is already available in Table S3.

**Response:** Please see the modification below:

Original text:

"For cooking, an OM/OC ratio of 1.4 was applied (Zhao et al., 2007). For vegetative detritus, OM/OC ratio of 2.1 was applied (Bae et al., 2006b). The OM/OC ratios for oxygenated OA were in the range of 1.85-2.3 (Zhang et al., 2005; Aiken et al., 2008), and the OM/OC ratio was 2.17 in secondary organic aerosols of PM$_{2.5}$ (Bae et al., 2006a). Therefore, an OM/OC ratio of 2.2 is applied in this study to convert the Other OC to OM."

Modified text:

"For cooking, vegetative detritus and secondary organic aerosols, OM/OC ratios were applied considering the low contribution of inorganic species to PM$_{2.5}$ mass from these sources (Zhao et al., 2007; Bae et al., 2006b). The OM/OC ratios for oxygenated OA were in the range of 1.85-2.3 (Zhang et al., 2005; Aiken et al., 2008), and the OM/OC ratio was 2.17 in secondary organic aerosols of PM$_{2.5}$ (Bae et al., 2006a). Therefore, an OM/OC ratio of 2.2 is applied in this study to convert the Other OC to OM."

Please see **lines 705-711** in the revised manuscript.

Comment 28 - Line 649. Specify what does the range after the value means. Is it a confidence interval, a standard deviation or something else? Does it include the uncertainty of the OC/PM$_{2.5}$ ratios?

**Response:** The value in the sentence "As shown in Table S4, PM$_{2.5}$ mass was well

explained by those sources which accounted for 91.9±24.1% and 99.0±19.1% of online PM$_{2.5}$ in winter and summer, respectively." represents "mean±SD of the daily values", which is added in Table S5 in the revised manuscript to avoid any confusion. Because the uncertainties of OC/PM$_{2.5}$ ratios were not known, they were not considered in the calculation. The possible uncertainties of using OC/PM$_{2.5}$ ratios have been discussed in the revised manuscript, which are also illustrated above.

Comment 29 - Line 652 and foll.: Since these results are in a table it is not necessary to list them here.

**Response:** Please see modifications below:

The original text:

"On average, the source contributions in winter ranked as SNA (30.5 µg m$^{-3}$, 34.1% of reconstructed PM$_{2.5}$ and hereafter), coal combustion (industrial & residential CC; 17.7 µg m$^{-3}$, 21.4%), Other OM (14.6 µg m$^{-3}$, 14.8%), biomass burning (8.9 µg m$^{-3}$, 11.0%), gasoline & diesel (5.6 µg m$^{-3}$, 7.5%), geological minerals (5.3 µg m$^{-3}$, 7.0%), cooking (3.1 µg m$^{-3}$, 3.9%) and vegetative detritus (0.2 µg m$^{-3}$, 0.3%); in summer these ranked as SNA (17.7 µg m$^{-3}$, 48.5%), other OM (8.0 µg m$^{-3}$, 18.3%), coal combustion (4.7 µg m$^{-3}$, 14.6%), geological minerals (3.5 µg m$^{-3}$, 10.4%), cooking (0.9 µg m$^{-3}$, 2.8%), gasoline & diesel (0.8 µg m$^{-3}$, 2.6%), biomass burning (0.8 µg m$^{-3}$, 2.1%) and vegetative detritus (0.2 µg m$^{-3}$, 0.7%)."

The revised text:

"On average, the source contributions in winter ranked as SNA > coal combustion > Other OM > biomass burning > gasoline & diesel > geological minerals > cooking > vegetative detritus; in summer these ranked as SNA > other OM > coal combustion > geological minerals > cooking > gasoline & diesel > biomass burning > vegetative detritus."

Please see **lines 729-733** in the revised manuscript.

Comment 30 - Figure 9: I suggest to replace the daily data with the overall averages (now in table S4).

**Response:** The daily data in original Fig. 9 was replaced with the overall averages (now in Fig. 10). The text is also modified as below:

"The daily PM$_{2.5}$ contribution estimates and seasonal average source contributions are provided in Fig. S7 and Fig. 10, respectively. Detailed data and their relative abundance in the reconstructed PM$_{2.5}$ are summarized in Table S5."

Please see **lines 758-760** in the revised manuscript for revised Fig. 10 and **lines 722-724** for revised text.

Comment 31 - Conclusions: this section is mandatory according to the instructions for authors of this journal

**Response:** We have changed the "Summary" to "Conclusion" and the added value of this study with respect to the previous knowledge on PM$_{2.5}$/OC sources in Beijing are

provided in **lines 778-788** in the revised manuscript, which is also copied below:

"Although the annual average PM$_{2.5}$ levels in Beijing reduced from 88 µg m$^{-3}$ in year 2013 to 58 µg m$^{-3}$ in year 2017 (Vu et al., 2019), and the deweathered concentration of PM$_1$ decreased by -38% in 2017 comparing to 2007 (Zhang et al., 2020), our CMB modelling results indicate that the coal combustion and biomass burning still remained the dominant primary OC sources in winter 2016 and summer2017, with road traffic ranked as the third highest. Cooking was a more significant source of OC than biomass burning at the urban site during summer. Compared to other CMB studies in Beijing, our study revealed an increase of the contributions from coal combustion, biomass burning and traffic to PM$_{2.5}$ in winter 2016 compared to winter 2000, while those in this study remained similar compared to winter 2013."

**Comments from Reviewer #2**

In the manuscript by Xu et al., the atmospheric fine particle samples from winter and summer at an urban site in Beijing were collected and analyzed. The source apportionment analysis from the Chemical mass balance (CMB) model shows comparable results with that of Positive Matrix Factorization (PMF) analysis of co-located Aerosol Mass Spectrometer (AMS). The paper provides useful scientific evidence to the source contributions in Beijing. However, there are still a few important issues that need to be clarified.

**Response:** Thanks for the reviewer's comments. Please see our point-by-point responses below.

Comment 1 - Major issues: One of the major issues in the paper is that the AMS analyzed the non-refractory fraction of PM$_1$, but the CMB method was applied to PM$_{2.5}$ samples. The difference in particle size segments is an important factor that could contribute significantly to the difference between the results from these two source apportionment methods. However, the authors didn't give any discussion about it. Especially, as shown in Figure 8, for the AMS-PMF derived OOA concentrations (in PM$_1$), which generally show slightly higher concentrations than other OC estimated by CMB methods (in PM$_{2.5}$). The authors have to give some reasonable explanations for this.

**Response:** We have provided more discussion regarding the size fractions of two methods, please see the modification below.

Original discussion:

[revised manuscript text omitted]

may be mainly attributable to variable OM/OC ratios." Please see **lines 321-336** in the revised manuscript.

Comment 3 - Minor suggestions: Page 3, line 63-64: "A better understanding of the sources of PM$_{2.5}$ in Beijing is essential to provide scientific evidence to control the PM$_{2.5}$ pollution." Such an expression is inaccurate.

**Response:**

The original sentence "A better understanding of the sources of PM$_{2.5}$ in Beijing is essential to provide scientific evidence to control the PM$_{2.5}$ pollution." has been revised as "A better understanding of PM$_{2.5}$ sources in Beijing is essential, as it can provide important scientific evidence to develop measures to control PM$_{2.5}$ pollution.". Please see **lines 63-65** in the revised manuscript.

Comment 4 - Page 3, the last paragraph: Please pay attention to the logical order of these sentences. For example, you have to introduce the CMB method was used to do the source apportionment first, then state how the source profiles were determined. Similar mistakes were also found in P5, line 143-146.

**Response:** We have modified the paragraph to follow the logic, please see our revision below.

The original paragraph:

"In this study, PM$_{2.5}$ samples were collected at an urban site of Beijing during winter and summer 2016-2017. OC, EC, PAHs, alkanes, hopanes, fatty acids and monosaccharide anhydrides were determined. To ensure that the source profiles used in the CMB model are representative, we mainly selected those studies which had been based in China: straw burning (Zhang et al., 2007b), wood burning (Wang et al., 2009), gasoline and diesel vehicles (Cai et al., 2017), industrial and residential coal combustion (Zhang et al., 2008), and cooking (Zhao et al., 2015). Source contributions of organic carbon were examined and quantified applying the CMB model based on the source profiles mentioned above. The results of this study are discussed and compared with the results from Aerosol Mass Spectrometer-PMF analysis (AMS-PMF) (Ulbrich et al., 2009; Elser et al., 2016) to improve our understanding of different sources of PM$_{2.5}$, especially for secondary organic aerosols."

The revised paragraph:

"In this study, PM$_{2.5}$ samples were collected at an urban site of Beijing in winter 2016 and summer 2017. OC, EC, PAHs, alkanes, hopanes, fatty acids and monosaccharide anhydrides in the PM$_{2.5}$ samples were determined, and applied in the CMB model for apportioning the organic carbon sources. To ensure that the source profiles used in the CMB model are representative, we mainly selected data which had been determined in China. The objectives of this study are: 1) to quantify the contributions of pollution sources to OC by applying a CMB model and compare them with those at a rural site of Beijing; 2) to compare the source apportionment results by CMB with those from Aerosol Mass Spectrometer-PMF analysis (AMS-PMF), to improve our understanding

of different sources of OC." Please see **lines 106-115** in the revised manuscript.

P5, line 143-146: "The uncertainties from duplicate analyses of filters were <10%. Replicate analyses were conducted once every ten samples. All sample results were corrected by the values obtained from field blanks, which were 0.40 and 0.01 μg m$^{-3}$ for OC and EC, respectively." has been revised as "Replicate analyses of OC and EC were conducted once every ten samples. The uncertainties from duplicate analyses of filters were <10%. All sample results were corrected by the values obtained from field blanks, which were 0.40 and 0.01 μg m$^{-3}$ for OC and EC, respectively." to follow the logic order of the sentences. Please see **lines 158-161** in the revised manuscript.

Comment 5 - P5, line 159-160: Please describe exactly how much is "A portion of the filters." So did the other analyses, such as inorganic components using Ion Chromatograph and ICP-MS in Section 2.2.3 (P6, second paragraph).

**Response:** Please see added information below.

Line 159-160: "A portion of the filters was extracted 3 times with dichloromethane/methanol (HPLC grade, v/v: 2:1) under ultrasonication for 10 minutes." has been change to "9 cm$^2$ of the quartz filters were extracted 3 times with dichloromethane/methanol (HPLC grade, v/v: 2:1) under ultrasonication for 10 minutes.". Please see **lines 172-174** in the revised manuscript.

"Half of the PTFE filter was extracted with 10 mL ultrapure water for the analysis of inorganic ions." has been added in the beginning of the second paragraph on P6. Please see **lines 191-192** in the revised manuscript.

"Other elements including V, Cr, Co, Mn, Ni, Cu, Zn, As, Sr, Cd, Sb, Ba and Pb were analyzed by Inductively-coupled plasma-mass spectrometer (ICP-MS), the detection limits of them were 1.32, 0.25, 0.04, 0.06, 2.05, 1.25, 1.22, 1.74, 0.02, 0.03, 0.11, 0.06 and 0.04 ng m$^{-3}$, respectively." has been revised as "Other elements including V, Cr, Co, Mn, Ni, Cu, Zn, As, Sr, Cd, Sb, Ba and Pb were analyzed by Inductively-coupled plasma-mass spectrometer (ICP-MS) after extraction of 1/2 PTFE filter by diluted acid mixture (HNO$_3$/HCl), and the detection limits of them were 1.32, 0.25, 0.04, 0.06, 2.05, 1.25, 1.22, 1.74, 0.02, 0.03, 0.11, 0.06 and 0.04 ng m$^{-3}$, respectively." Please see **lines 199-203** in the revised manuscript.

Comment 6 - P7, First paragraph: Please rephrase and reorganized these sentences.

**Response:** Please find the modifications below.

The original paragraph:

[revised manuscript text omitted]

Comment 8 - P18, line 560-565, and P7 line 245-251: Unnecessary to repeat this.

**Response:** To avoid repetition the text in P7 line 245-251 in the original manuscript "The PMF analysis resulted in an optimal solution of 2 primary factors in summer: traffic-related hydrocarbon-like OA (HOA) and cooking OA (COA) and 3 secondary factors of oxygenated OA (OOA): OOA1, OOA2, OOA3. In winter, 3 primary factors were identified: coal combustion OA (CCOA), COA, biomass burning OA (BBOA), and 3 secondary factors: oxidized primary OA (OPOA), less-oxidized OA (LOOOA), and more-oxidized OA (MOOOA)." has been deleted.

Comment 9 - P19-P20: when describing the comparisons between the results from CMB and AMSPMF, the language and symbols sometimes are quite confusing; please reorganize them.

**Response:** The phrases of AMS-PMF and CMB resolved OC sources are revised as suggested to avoid any confusion:

[revised manuscript text omitted]

---

## Author Response (AR2)

**Point-by-Point Response to Reviewers' Comments**

Manuscript Ref: acp-2020-1020

Title: Source Apportionment of Fine Aerosol at an Urban Site of Beijing using a Chemical Mass Balance Model

Journal: Atmospheric Chemistry and Physics

**Further comment from Reviewer**

Page3 last paragraph Line 90-98: The authors should mention that the good performance of CMB and its comparability with other receptor modelling techniques was demonstrated in intercomparison exercises.

**Response:** As suggested by the reviewer, this is now added in the revised manuscript.

The original paragrapgh "Chemical Mass balance (CMB) model has been used for source apportionment of PM worldwide, including in the US (Antony Chen et al., 2010), UK (Yin et al., 2015), and China (Chen et al., 2015b). The CMB model assumes that source profiles remain unchanged between the emitter and receptor (Sarnat et al., 2008; Viana et al., 2008). Xu et al. (2021) compared the source apportionment results of fine particles by multiple receptor modelling approaches, and found that CMB can provide the most complete and representative source apportionment of Beijing aerosols." has been revised as

"Chemical Mass balance (CMB) model has been used for source apportionment of PM worldwide, including in the US (Antony Chen et al., 2010), UK (Yin et al., 2015), and China (Chen et al., 2015b). The CMB model assumes that source profiles remain unchanged between the emitter and receptor (Sarnat et al., 2008; Viana et al., 2008). The good performance of CMB and its comparability with other receptor modelling techniques was demonstrated in an intercomparison exercise conducted in Beijing (Xu et al., 2021)."

Please see **lines 90-96** in the revised manuscript.